# Scaling Instruction-Tuned LLMs to Million-Token Contexts via Hierarchical Synthetic Data Generation

**Linda He[1]\* Jue Wang[2] Maurice Weber[2] Shang Zhu[2] Ben Athiwaratkun[2] Ce Zhang[2,3]**
[1]Harvard University   [2]Together AI   [3]University of Chicago
lindahe@college.harvard.edu, {jue, maurice, shang, ben}@together.ai,
cez@uchicago.edu

## Abstract

Large Language Models (LLMs) struggle with long-context reasoning, not only due to the quadratic scaling of computational complexity with sequence length but also because of the scarcity and expense of annotating long-context data. There has been barely any open-source work that systematically ablates long-context data, nor is there any openly available instruction tuning dataset with contexts surpassing 100K tokens. To bridge this gap, we introduce a novel post-training synthetic data generation strategy designed to efficiently extend the context window of LLMs while preserving their general task performance. Our approach scalably extends to arbitrarily long context lengths, unconstrained by the length of available real-world data, which effectively addresses the scarcity of raw long-context data. Through a step-by-step rotary position embedding (RoPE) scaling training strategy, we demonstrate that our model, with a context length of up to 1M tokens, performs well on the RULER benchmark and InfiniteBench and maintains robust performance on general language tasks.

## 1 Introduction

The capabilities of Large Language Models (LLMs) have significantly advanced, enabling impressive performance across a wide range of natural language processing tasks (Wu et al., 2023; Jiang et al., 2023; Wei et al., 2022). However, managing long contexts remains a major challenge, which limits the practical utility of LLMs in tasks such as document comprehension and summarization, code generation, lifelong conversations, and complex agent scenarios (Liu et al., 2023; Meng et al., 2023). Extending context lengths to 1M tokens marks a critical breakthrough for applications requiring processing beyond a 128K token limit. For instance, company-wide document retrieval benefits from efficiently analyzing extensive organizational histories stored in unstructured formats, while interconnected project timelines and legal documents gain from enhanced reasoning across multi-document datasets.

To extend the context length of LLMs, current approaches focus on either architectural innovations like efficient attention mechanisms (Katharopoulos et al., 2020; Gu & Dao, 2024) or scaling positional embeddings (Chen et al., 2023; Peng et al., 2023) and continual pretraining on natural long-form data, such as books and web data. However, the RULER benchmark (Hsieh et al., 2024) shows that many models struggle to maintain consistent performance as context length increases, even when claiming to support longer contexts. This highlights the need for high-quality instruction data to fully utilize the nuances of long-form content. Acquiring such data is challenging and costly, as open-source datasets often fall short in document length, relevance, and tasks requiring genuine long-range understanding. To date, no open-source instruction-tuning datasets exceed 100K tokens, creating a significant gap between theoretical and practical long-context capabilities of LLMs (Li et al., 2024; Zhao et al., 2024).

To address limitations in extending LLM context length, we propose an effective long-context instruction data generation pipeline, as illustrated in Figure 1. Our pipeline leverages short-context

---

\*Work done during an internship at Together AI.

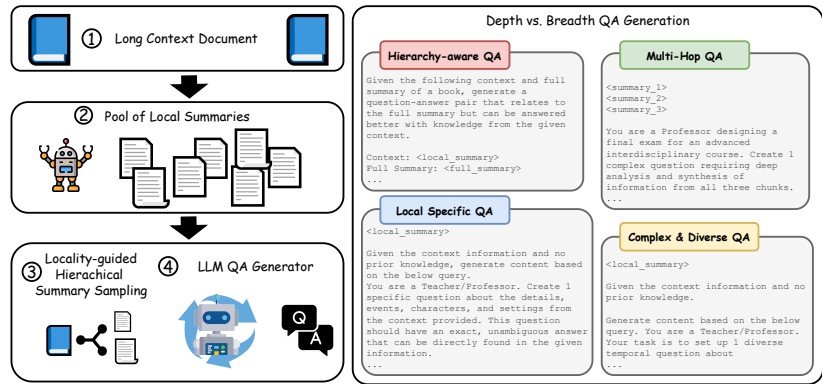

Figure 1: High-level overview over our approach to automatically generate QA pairs for long context documents. (1) In the first step, we split a document into small and medium chunks which are then (2) summarized by an off-the-shelf LLM requiring only smaller context windows. In (3) we sample summaries at different localities in a hierarchical manner, balancing local and global views of the original document. In (4) we generate questions based on the sampled summaries. In the right panel, we show a subset of prompts used to generate diverse and complex questions, given the sampled summaries.

models to create long-context instruction data using three key methods: (a) *Hierarchical question ordering*: structuring questions in a logical sequence to ensure coherent reasoning across contexts; (b) *Diverse question type pool*: maintaining a wide range of question types, including hierarchical-aware, multi-hop, local-specific, and other complex types to handle varied tasks; and (c) *Multi-document integration*: incorporating multiple documents to generate data with arbitrary context lengths. The contributions of this paper are threefold:

1. **Extensive and scalable long-context data generation strategy**: We present, to the best of our knowledge, the first extensive strategy for synthetically generating long-context data with comprehensive ablation tests and evaluations. Our highly scalable approach is unconstrained by the length of available real-world data, effectively combining multiple documents with diverse, complex questions. This hierarchical method ensures logical coherence and sequence integrity.
2. **Extensive evaluation of core strategies**: We conduct extensive evaluations on shorter context lengths (100K and 180K) to demonstrate the effectiveness of our hierarchical strategy, multi-document combinations, and diverse question-answer pair generation. These evaluations validate that our core strategies work well across various tasks and context lengths.
3. **Scaling to 1M context length**: We successfully extend LLaMA-3.1-8B-Instruct to a context length of 1 million tokens. Our model significantly outperforms the LLaMA-3.1-8B-Instruct model in zero-shot RoPE scaling to a 1M context window on the RULER benchmark and surpasses the gradientai/Llama-3-8B-Instruct-Gradient-1048k model trained by Gradient AI. Additionally, our model outcompetes LLaMA-3.1-8B-Instruct on InfiniteBench while maintaining strong performance on LongBench and MMLU.

The remainder of this work is organized as follows. In Section 2 we place our work in the landscape of existing literature around methods to address long context capabilities of LLMs. Section 3 presents our method for generating long-context instruction tuning data. Our approach is then validated in Section 4 with a series of extensive and representative experiments. Finally, we conclude in Section 5.

## 2 RELATED WORK

Adapting transformers to enable longer context capabilities is a critical area of research in natural language processing. This effort primarily focuses on three key directions: (1) architectural modifications to the transformer model itself, (2) improvements in positional embedding techniques, and (3) the development and utilization of more extensive long-context datasets.

**Efficient Attention Mechanisms.** To address the quadratic computational and memory demands of standard transformer self-attention, researchers have developed various architectural modifica-

tions to improve efficiency and extend context lengths. Notable examples include Longformer (Beltagy et al., 2020), which combines sliding window and global attention, and BlockTransformer (Ho et al., 2024), which employs hierarchical global-to-local modeling. Linear Attention methods (Katharopoulos et al., 2020) reformulate self-attention for linear complexity, while InfiniteTransformer (Munkhdalai et al., 2024) incorporates unbounded long-term memory through continuous-space attention. State space models like Mamba (Gu & Dao, 2024) capture long-range dependencies efficiently without explicit attention mechanisms. Despite these advancements, bridging the gap with high-quality data remains a critical challenge and is the focus of this work.

**Position Embedding Extension.** Advances in positional encoding methods have enabled language models to handle longer sequences effectively. Techniques like RoPE (Su et al., 2023), ALiBi (Press et al., 2022), and xPos (Sun et al., 2022) have emerged as prominent solutions. RoPE has gained widespread adoption in LLaMA (Touvron et al., 2023), b) and PaLM (Anil et al., 2023), due to its ability to represent relative positions and its theoretical grounding in the complex plane. A breakthrough showed that RoPE's embeddings could extend to longer contexts with minimal or no fine-tuning (Men et al., 2024), leading to two key approaches: Positional Interpolation (PI) (Chen et al., 2023) which linearly scales positional indices to extend context length, and NTK-aware Scaling RoPE (Peng et al., 2023) which combines high-frequency extrapolation with low-frequency interpolation. While these developments improve model performance with longer inputs, they rely heavily on limited long-context data for fine-tuning.

**Long Context Data.** Recent work, such as LongT5 (Guo et al., 2022) and LongAlpaca (Chen et al., 2024), has shown the benefits of additional pretraining on long sequences, enabling models to better capture extended context. Methods like combining multiple short-context sequences (Xiong et al., 2023) have also emerged as promising ways to efficiently extend context lengths. However, a significant gap remains in generating high-quality instruction-tuning data exceeding 100K context lengths. Few open-source efforts address this need. Our work introduces a scalable pipeline for generating long-context instruction-tuning data by systematically combining multiple documents, diverse questions, and a hierarchical strategy to ensure coherence and structure.

**Synthetic Data Generation.** Synthetic data generation offers a promising path for scaling language models across diverse tasks and complex instructions. AutoEvol-Instruct (Zeng et al., 2024), automates the evolution of instruction datasets using large language models, reducing the need for extensive human intervention. WizardLM (Xu et al., 2023) employs Evol-Instruct to iteratively evolve and scale instruction complexity, achieving strong results on benchmarks like MT-Bench and Vicuna's evaluation set. Auto Evol-Instruct (Zeng et al., 2024) further refines this process with an iterative evolution strategy, while Self-Instruct (Wang et al., 2023) enhances instruction-following performance through data synthesis. Our work extends this research by generating long-context data tailored for instruction tuning.

## 3 METHOD

In this section, we describe our methodology for generating coherent instructions from a single document and scaling it to multiple documents to curate long-context datasets beyond the context length of available raw data. Section 3.1 outlines our strategy for ensuring (1) *quality and complexity* and (2) *coherent ordering* of generated question-answer pairs. Section 3.2 expands on scaling to longer context lengths using multiple documents. Figure 1 provides an overview of our long-context synthetic data generation pipeline.

### 3.1 COHERENT INSTRUCTIONS FROM A SINGLE DOCUMENT

The quality of long-context instruction-tuning datasets is driven by two key factors: (1) the *complexity and diversity* of the generated instructions, and (2) the *structured ordering* of questions and instructions. To address these, we devised a bifurcated strategy targeting each component.

**Quality, Diversity, and Complexity of Instructions.** As illustrated in Figure 1, our methodology for generating rich, diverse, and complex instructions leverages the key insight that *short-context models can be used to generate long-context instruction-tuning data*. The core approach involves dividing the input document into smaller chunks, typically 4K tokens, enabling models optimized for shorter contexts to process these segments with *greater precision and clarity*. We curate an initial set of prompts covering multiple dimensions of instruction complexity, such as temporal reasoning, thematic inquiry, and character-based scenarios (full set in Appendix B). During question-

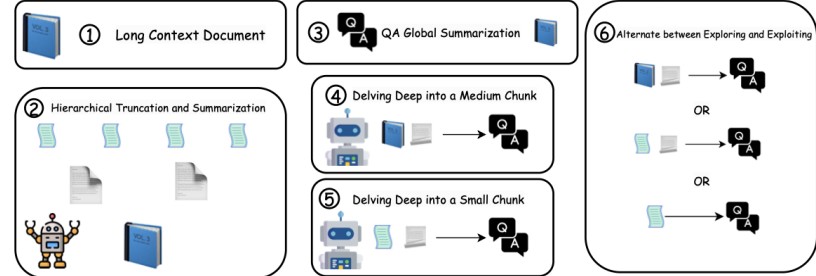

Figure 2: High-level overview over our approach to generate order-following QAs. (1) Input a raw long context document. (2) Split the document into small, medium, and global chunks, and generate summaries at each level. (3) The first QA is based on the global summary. (4) We randomly select a medium chunk to generate a QA, (5) then delve deeper by selecting a small chunk within it for another QA. (6) To continue, the process alternates between exploiting the same small chunk or exploring new medium or small chunks to generate further QAs.

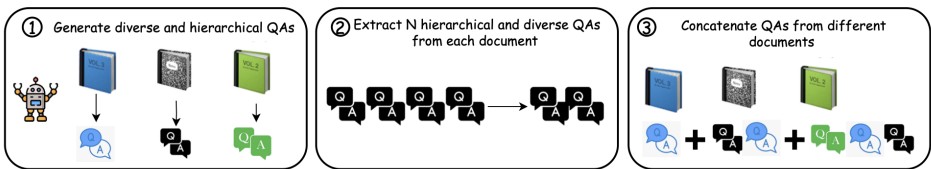

Figure 3: High-level overview over our approach to curate long context data using multiple documents. (1) Diverse and hierarchical QAs are generated at different levels of granularity for each document. (2) $N$ hierarchical and diverse QAs are sampled and extracted from each document. (3) QAs from different documents are combined, maintaining a balance of hierarchical and diverse questions across the entire set. $N = 5$ in our algorithm, and when we revisit previous documents in step (3), we sample 3 hierarchial questions for each document with 60 % probability as well as 9 total diverse questions from all previous documents.

answer pair generation, a small chunk and one question are randomly selected to generate a pair. To ensure broader contextual understanding, we incorporate multi-hop questions spanning 2–4 chunks, enabling cross-chunk question-answer pairs.

**Ensuring Coherent Order.** To ensure logical and coherent QA generation, we use a hierarchical strategy to split, summarize, and generate questions from long documents (see Figure 2), balancing exploration and exploitation. The document is first divided into large sections of 12K tokens, then into smaller 4K-token chunks linked hierarchically to connect broader and granular segments. The first QA is based on the global summary to give a high-level overview of the document. Then, we randomly select a medium chunk to generate a QA, and then delve deeper by selecting a small chunk within it for another QA. To continue, the process alternates between exploiting the same small chunk or exploring new medium or small chunks to generate further QAs. This iterative process ensures a balance between specificity and diversity, covering both localized details and broader document sections. The hierarchical structure ensures logical progression from broad QAs to detailed ones. The detailed algorithm and pseudocode are provided in Appendix A.

### 3.2 EXTENDING TO LONGER CONTEXT LENGTHS USING MULTIPLE DOCUMENTS

Here we extend our methodology to handle longer contexts by concatenating multiple documents and generating coherent hierarchical and diverse QA pairs across them. The workflow is visualized in Figure 3 and the detailed algorithm is provided in Appendix A. Below, we clearly define the parameters $N_1$, $N_2$, and $N_3$, which govern the selection of hierarchical and diverse QA pairs, ensuring logical continuity and broad reasoning across documents. For each document, the process proceeds as follows:

1. $N_1$ hierarchical QA pairs and $N_1$ diverse QA pairs: After processing each document, $N_1 = 5$ hierarchical follow-up questions are added. These questions are designed to capture contextually related information within the document, creating a logical order of reasoning and flow across sections. Moreover, another $N_1 = 5$ diverse QA pairs for this document is added as well, designed to capture specific details of the document.

2. $N_2$ diverse QA pairs: Next, $N_2 = 9$ diverse QA pairs are added. These questions are sampled from all previously visited documents where diverse QA pairs have not already been sampled. This approach ensures cross-referencing between documents.

3. $N_3$ revisiting hierarchical QA pairs: For every previously visited document, there is a 60% probability of sampling $N_3 = 3$ hierarchical follow-up questions. These are added to revisit earlier contexts, fostering a richer and interconnected understanding of the content.

This process is repeated iteratively for all $K$ documents in the dataset to create a comprehensive instruction-tuning dataset that balances within-document reasoning, cross-document relationships, and revisiting earlier content for contextual continuity. We also present an example of a concatenated data example in Appendix C.

## 4 EXPERIMENTS

In this section, we validate our long-context data generation approach through a series of experiments. In Section 4.2, we extend LLaMA-3.1-8B-Instruct to a 1M context-length model using stepwise RoPE scaling and hierarchical, diverse QA data generated by Qwen-2-72B. Our 1M model delivers excellent results on ultra-long contexts while maintaining strong performance on short and medium-length contexts. In Section 4.3, we evaluate robustness using smaller and same-sized generator models (Qwen-2.5-7B and LLaMA-3.1-8B-Instruct), confirming our models achieve strong performance across ultra-long, short, and medium contexts. These findings highlight the scalability and effectiveness of our approach across generator model sizes. In Section 4.4, we present ablation studies showing how our hierarchical strategy and diversified questions significantly improve long-context instruction tuning, focusing on 180K with two documents.

### 4.1 SETUP

**Models.** We use LLaMA-3.1-8B-Instruct as the base model for instruction-tuning, given its capability as a leading open-source LLM. To validate robustness, we employ various generator models for synthetic data: Qwen-2-72B-Instruct (large, high-quality data), Qwen-2.5-7B-Instruct (smaller), and LLaMA-3.1-8B-Instruct (same size). This demonstrates that our improvements are not reliant on very large models and that smaller models can achieve similar gains. We also benchmark against the Gradient AI model (gradientai/Llama-3-8B-Instruct-Gradient-1048k), a 1M context-length model trained on 1.4 billion tokens, showing that our method outperforms existing baselines.

**Hardware.** We fine-tuned our models on a SLURM cluster using 8 to 32 H100 GPUs across up to 4 nodes, connected via InfiniBand for efficient multinode training. We used FSDP to shard the model across GPUs and implemented DeepSpeed Ulysses sequence parallelism for long-context training.

**Datasets.** Our primary dataset is the Together long books dataset[1], processed into approximately **1.4 billion** tokens, distributed across these stages: 2000 samples of 180K tokens, 1280 samples of 350K tokens, 600 samples of 650K tokens, and 200 samples of 1M tokens. We generated 582,900 QA pairs with hierarchical and diverse questions for robust instruction-tuning using the Together AI inference API [2]. By sending 32 simultaneous API requests, it took about two days to create our full long-context instruction dataset, comprising 7,772 books. For each book, we generated 25 hierarchical and 50 diverse questions, resulting in 582,900 QA pairs alongside global summaries. During training, we calculate loss solely on answers, masking out questions and context to ensure the model focuses on reasoning and generating accurate answers without being penalized for reproducing input content.

**Evaluation Protocol.** We evaluated our models using: 1) **InfiniteBench** (Zhang et al., 2024): Designed for LLMs on extended contexts, it includes tasks like key-value retrieval, summarization, and QA on data exceeding 100K tokens. We evaluated the first 150 samples per task, excluding coding tasks as our data lacks code. 2) **LongBench** (Bai et al., 2024): Focused on medium-context tasks (10K tokens), it assesses summarization, QA, and fact-checking across multiple domains, offering a measure of general capabilities. We excluded coding tasks. 3) **RULER** (Hsieh et al., 2024): RULER is a synthetic benchmark designed to evaluate how well models handle complex, real-world tasks in long contexts. Unlike traditional retrieval-based tasks like Needle-in-a-Haystack (NIAH), which focus on extracting specific pieces of information from distractor texts, RULER tests models'

---

[1] https://huggingface.co/datasets/togethercomputer/Long-Data-Collections
[2] https://api.together.xyz/

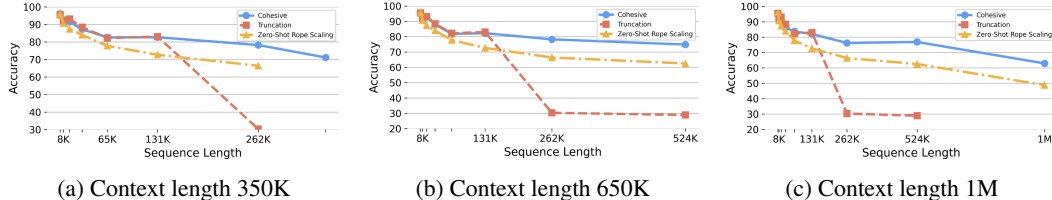

| (a) Context length 350K | (b) Context length 650K | (c) Context length 1M |
|---|---|---|

Figure 4: Effective context length up 1M tokens using Qwen-2-72B-Instruct as generator on RULER.

Table 1: Model performance on InfiniteBench (100K tokens) using Qwen-2-72B-Instruct as generator.

| Metric | LLaMA-3.1-8B-Instruct | gradient-ai-model | 180K | 350K | 650K | 1M |
|---|---|---|---|---|---|---|
| Retrieve.PassKey | **100.00** | **100.00** | **100.00** | **100.00** | **100.00** | **100.00** |
| Retrieve.Number | 95.33 | 99.83 | 99.33 | **100.00** | **100.00** | **100.00** |
| Retrieve.KV | 42.66 | 15.60 | 88.66 | **92.00** | 63.33 | 57.33 |
| En.Sum | **27.63** | 17.02 | 24.01 | 23.51 | 23.68 | 23.06 |
| En.QA | 24.83 | 14.31 | **34.26** | 33.23 | 31.72 | 31.97 |
| En.MC | 68.00 | 57.20 | 74.00 | 72.00 | **75.33** | 74.00 |
| En.Dia | 16.66 | 5.00 | 18.00 | 18.00 | **22.00** | 16.00 |
| Math.Find | 35.33 | 19.42 | **37.33** | 35.33 | 36.00 | 36.00 |
| **Average** | 51.31 | 41.04 | **59.45** | 59.26 | 56.51 | 54.80 |

ability to comprehend deeper relationships and manage long-range dependencies. Given a specified context length, RULER generates synthetic tasks across multiple categories, including multi-hop reasoning and document tracing, and measures the model's accuracy. In our evaluation, we sampled 130 tasks for each context length across 13 categories, totaling over 150 million tokens. 4) **MMLU** (Hendrycks et al., 2021): This benchmark evaluates general model performance across multiple domains, assessing both breadth and depth of understanding. It includes tasks spanning STEM, humanities, and social sciences, with varying difficulty levels. MMLU ensures that improvements in handling long-context tasks do not cause regression in overall model capabilities.

## 4.2 MAIN RESULTS: SCALING UP TO LONGER CONTEXT LENGTHS (350K, 650K, 1M)

To extend Llama-3.1-8B-Instruct to a 1M context model, we applied stepwise rope scaling. Training started with 180K tokens and progressed through checkpoints at 350K, 650K, and 1M tokens, concatenating 4, 8, and 12 documents as per the algorithm in Section 3.2. We compiled 2000 samples at 180K, 1280 at 350K, 600 at 650K, and 200 at 1M tokens. Data was generated using Qwen-2-72B, fine-tuned on Llama-3.1-8B-Instruct with rope scaling at a 6e-5 learning rate for 1 epoch. Training the 650K model took 30 hours, and the 1M model took an additional 52.5 hours.

An earlier ablation test combining two documents (Section 4.4) showed that combining hierarchical and diverse questions with a fixed number of QAs and global summarization is optimal for handling long contexts. We extended this setup for ultra-long context data, with each document followed by $N_1 = 5$ hierarchical and $N_1 = 5$ diverse questions. When revisiting previous documents, there is a 60% chance of extracting $N_2 = 3$ hierarchical question from each document and $N_3 = 9$ diverse questions sampled from all prior documents.

Figure 4 shows the effective context lengths of the 350K, 650K, and 1M models on the RULER benchmark. For comparison, we performed zero-shot rope scaling on the LLaMA-3.1-8B-Instruct model and included results using input truncation for context lengths above 128K as an additional baseline. On contexts shorter than 128K, our models performed comparably to LLaMA-3.1-8B-Instruct and surpassed it with zero-shot rope scaling. This demonstrates the robustness of our models on short and medium contexts. For contexts longer than 128K, our models significantly outperformed both baselines, with their strengths becoming more evident as context length increased. Raw evaluation results are in Appendix D.

Table 2: Model performance on LongBench (10K tokens) using Qwen-2-72B-Instruct as generator.

| | LLaMA-3.1-8B-Instruct | Gradient-AI-Model | 180K | 350K | 650K | 1M |
|---|---|---|---|---|---|---|
| Single Document | **46.91** | 30.71 | 45.83 | 45.88 | 45.24 | 45.15 |
| Multi-Document | 41.45 | 12.45 | 41.71 | **41.75** | 41.13 | 41.29 |
| Summarization | **26.10** | 21.72 | 25.14 | 24.97 | 24.26 | 24.98 |
| Few-shot Learning | **63.48** | 59.69 | 62.22 | 61.66 | 60.00 | 59.27 |
| Synthetic Tasks | 67.48 | 55.50 | **68.17** | 67.50 | 65.00 | 66.42 |
| All | **48.11** | 35.89 | 47.58 | 47.34 | 46.18 | 46.42 |

Table 3: Model performance on MMLU using Qwen-2-72B-Instruct as the generator.

| Category | LLaMA-3.1-8B-Instruct | gradient-ai-model | 350K-model | 650K-model | 1M-model |
|---|---|---|---|---|---|
| mmlu | $68.21 \pm 0.37$ | $60.48 \pm 0.39$ | $66.29 \pm 0.38$ | $65.80 \pm 0.38$ | $65.08 \pm 0.38$ |
| humanities | $64.23 \pm 0.67$ | $55.75 \pm 0.69$ | $61.51 \pm 0.68$ | $61.02 \pm 0.68$ | $61.02 \pm 0.68$ |
| other | $73.03 \pm 0.77$ | $67.04 \pm 0.82$ | $72.84 \pm 0.77$ | $71.84 \pm 0.78$ | $71.84 \pm 0.78$ |
| social sciences | $77.48 \pm 0.74$ | $70.46 \pm 0.80$ | $76.81 \pm 0.74$ | $75.27 \pm 0.76$ | $75.27 \pm 0.76$ |
| stem | $60.36 \pm 0.83$ | $51.32 \pm 0.86$ | $59.44 \pm 0.84$ | $57.72 \pm 0.84$ | $57.72 \pm 0.84$ |

To further validate our approach, we compared it to the Gradient AI model (gradientai/Llama-3-8B-Instruct-Gradient-1048k), a 1M context model, on InfiniteBench, LongBench, and MMLU benchmarks. Table 1 compares models across context lengths on InfiniteBench, while Table 2 focuses on LongBench. All our models (180K, 350K, 650K, 1M) consistently outperforms the Gradient AI model on InfiniteBench, showcasing the effectiveness of our hierarchical, diversified QA-based data-generation strategy. The 180K and 350K models scored 59.45 and 59.26, significantly exceeding the LLaMA-3.1-8B-Instruct baseline of 51.31. The 650K model scored 56.51, and the 1M model achieved a strong 54.80. [3]

Notably, while the Retrieve.KV task shows the most significant improvement, tasks like Retrieve.Number, En.MC, and Math.Find also display meaningful gains. The improvement on Retrieve.KV stems from our data-generation methodology, which uses a structured mix of hierarchical and diverse questions while revisiting prior documents. This encourages the model to associate relevant sections, aligning with the demands of key-value retrieval and RAG techniques, where accurate context memorization is critical. Beyond key-value retrieval, our model excels on other tasks: on En.MC, the 650K model scored 75.33, surpassing the baseline (68.00) and Gradient AI model (57.20). On Math.Find, it scored 36.00 at 650K, outperforming the Gradient AI model (19.42), showcasing improved reasoning capabilities.

As shown in Table 2, , our models maintain robust short-context performance on LongBench, despite being trained for significantly longer contexts (up to 1M tokens). For example, our 1M context-length model achieves an average score of 46.42, comparable to the baseline LLaMA-3.1-8B-Instruct model (48.11). This demonstrates that while optimized for ultra-long contexts, the model generalizes effectively to shorter contexts, such as those on LongBench. Minor regressions in tasks like summarization are due to trade-offs when training for extended contexts. As the model adapts to handle extremely long contexts, small task-specific adjustments may impact short-context performance. However, these regressions are minimal and expected, given the differences between short- and long-context tasks. Despite these trade-offs, our model consistently outperforms the Gradient AI model (35.89) on all LongBench tasks, demonstrating the effectiveness of our hierarchical and diversified instruction-tuning approach.

As detailed in Table 3, our model demonstrated minimal regression in general task performance despite significant improvements in ultra-long-context tasks. For instance, our model retained com-

---

[3]The results dropped likely due to multi-node training, as we believe our 650K and 1M models are undertrained because of the extended time required to train and the communication overhead from NCCL.

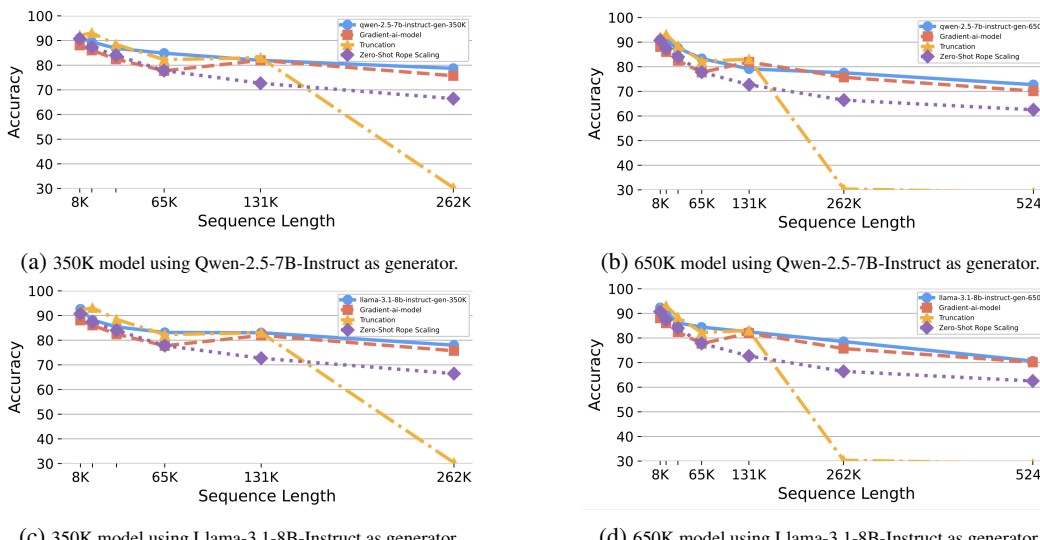

(a) 350K model using Qwen-2.5-7B-Instruct as generator.

(b) 650K model using Qwen-2.5-7B-Instruct as generator.

(c) 350K model using Llama-3.1-8B-Instruct as generator.

(d) 650K model using Llama-3.1-8B-Instruct as generator.

Figure 5: Effective context length using Llama-3.1-8B-Instruct and Qwen-2.5-7B-Instruct as generators on RULER.

Table 4: InfiniteBench performance with Llama-3.1-8B-Instruct and Qwen-2.5-7B-Instruct as generators.

| Task | LLaMA-3.1-8B-Instruct | gradient-ai-model | 180K-llama-gen | 350K-llama-gen | 650K-llama-gen | 180K-qwen-gen | 350K-qwen-gen | 650K-qwen-gen |
|---|---|---|---|---|---|---|---|---|
| Retrieve.PassKey | 100.00 | 100.00 | 100.00 | 100.00 | 100.00 | 100.00 | 100.00 | 100.00 |
| Retrieve.Number | 95.33 | 99.33 | 99.04 | 100.00 | 100.00 | 99.76 | 100.00 | 100.00 |
| Retrieve.KV | 42.66 | 13.33 | 85.47 | 89.33 | 42.14 | 89.52 | 85.33 | 52.66 |
| En.Sum | 27.63 | 17.02 | 25.68 | 26.85 | 26.64 | 26.97 | 27.70 | 26.74 |
| En.QA | 24.83 | 15.84 | 33.39 | 35.67 | 33.37 | 32.30 | 29.55 | 29.67 |
| En.MC | 68.00 | 61.33 | 58.00 | 60.66 | 66.00 | 63.33 | 61.33 | 64.66 |
| En.Dia | 16.66 | 4.00 | 19.50 | 14.66 | 20.00 | 27.33 | 21.33 | 23.33 |
| Math.Find | 35.33 | 26.66 | 36.66 | 32.66 | 35.33 | 30.00 | 34.66 | 38.00 |
| **Average** | 51.31 | 42.19 | 57.22 | 57.48 | 52.94 | 58.65 | 57.49 | 54.38 |

petitive MMLU scores (e.g., 68.21 ± 0.37 for the baseline and 65.08 ± 0.38 for the 1M model), whereas the Gradient AI model showed marked degradation on both MMLU and LongBench. This reinforces the robustness of our method, ensuring that gains in ultra-long-context performance do not compromise broader capabilities. In conclusion, our models excel at ultra-long-context tasks on RULER and InfiniteBench, outperforming the base LLaMA-3.1-8B-Instruct and Gradient AI models while maintaining strong performance on general tasks like MMLU and LongBench.

## 4.3 Validating Robustness Across Generator Models

To validate that observed improvements are not solely due to using a large generator model (e.g., Qwen-2-72B), we trained and evaluated models with Qwen-2.5-7B and LLaMA-3.1-8B-Instruct as generators. By employing smaller or similarly sized models, we demonstrated the robustness and generalizability of our hierarchical QA data-generation strategy. Additionally, we benchmarked against the Gradient AI model (gradientai/Llama-3-8B-Instruct-Gradient-1048k), a 1M context model trained on 1.4 billion tokens. While our models were trained only up to 650K tokens to validate the approach, the same method can seamlessly scale to 1M tokens. Our models outperformed the Gradient AI baseline across all long-context benchmarks, achieving higher accuracy on InfiniteBench and RULER, while preserving general task performance on MMLU and LongBench.

Figure 5 highlights effective context length using Llama-3.1-8B-Instruct and Qwen-2.5-7B as generators on RULER. On all settings (350K, 650K), our hierarchical approach outperformed the Gradient AI model and the zero-shot baselines across context lengths. Table 4 summarizes results on InfiniteBench (100K context length). Our approach again consistently outperformed both the base LLaMA-3.1-8B-Instruct model and the Gradient AI model. This demonstrates that even smaller generator models produce high-quality data for instruction-tuning.

Table 5: LongBench performance with Llama-3.1-8B-Instruct and Qwen-2.5-7B-Instruct as generators.

| Task | LLaMA-3.1-8B-Instruct | gradient-ai-model | 180K-llama-gen | 350K-llama-gen | 650K-llama-gen | 180K-qwen-gen | 350K-qwen-gen | 650K-qwen-gen |
|---|---|---|---|---|---|---|---|---|
| single-document | 46.91 | 30.75 | 46.48 | 46.64 | 46.53 | 46.20 | 46.70 | 46.28 |
| multi-document | 41.45 | 12.45 | 38.69 | 38.75 | 37.54 | 40.76 | 41.90 | 39.31 |
| summarization | 26.10 | 21.72 | 25.28 | 25.10 | 24.68 | 25.05 | 24.83 | 24.90 |
| few-shot learning | 63.48 | 59.70 | 61.56 | 62.79 | 60.50 | 61.92 | 61.56 | 60.69 |
| synthetic tasks | 67.48 | 55.50 | 66.17 | 67.75 | 66.00 | 67.11 | 67.60 | 67.10 |
| **Average** | 48.11 | 35.89 | 47.23 | 47.72 | 46.20 | 47.95 | 47.97 | 47.00 |

Table 6: MMLU performance with Llama-3.1-8B-Instruct and Qwen-2.5-7B-Instruct as generators.

| Category | LLaMA-3.1-8B-Instruct | gradient-ai-model | 180K-llama-gen | 350K-llama-gen | 650K-llama-gen | 180K-qwen-gen | 350K-qwen-gen | 650K-qwen-gen |
|---|---|---|---|---|---|---|---|---|
| mmlu | $68.21 \pm 0.37$ | $60.48 \pm 0.39$ | $66.99 \pm 0.38$ | $66.74 \pm 0.38$ | $65.93 \pm 0.38$ | $67.33 \pm 0.38$ | $65.78 \pm 0.38$ | $64.60 \pm 0.38$ |
| humanities | $64.23 \pm 0.67$ | $55.75 \pm 0.69$ | $62.32 \pm 0.67$ | $61.38 \pm 0.68$ | $60.57 \pm 0.68$ | $62.81 \pm 0.67$ | $59.68 \pm 0.68$ | $59.45 \pm 0.68$ |
| other | $73.03 \pm 0.77$ | $67.04 \pm 0.82$ | $72.90 \pm 0.77$ | $73.03 \pm 0.76$ | $72.87 \pm 0.76$ | $73.51 \pm 0.76$ | $73.00 \pm 0.76$ | $73.45 \pm 0.77$ |
| social sciences | $77.48 \pm 0.74$ | $70.46 \pm 0.80$ | $76.70 \pm 0.74$ | $76.93 \pm 0.74$ | $75.53 \pm 0.75$ | $76.76 \pm 0.74$ | $75.66 \pm 0.75$ | $71.87 \pm 0.77$ |
| stem | $60.36 \pm 0.83$ | $51.32 \pm 0.86$ | $58.67 \pm 0.84$ | $58.61 \pm 0.84$ | $57.72 \pm 0.84$ | $58.77 \pm 0.84$ | $58.14 \pm 0.84$ | $56.49 \pm 0.85$ |

Table 5 evaluates model performance on LongBench (10K context length). Despite being optimized for ultra-long contexts, our approach retains strong performance on shorter contexts, comparable to LLaMA-3.1-8B-Instruct. For example, with Qwen-2.5-7B-Instruct as the generator, our model scored 47.00 at 650K, closely matching LLaMA-3.1-8B-Instruct's 48.11. Our model also outperforms Gradient AI (35.89) across all LongBench tasks. Table 6 shows our models' minimal regression in MMLU performance. The 650K trained using LLaMA-3.1-8B-Instruct as generator scored $65.93 \pm 0.38$, close to LLaMA-3.1-8B-Instruct ($68.21 \pm 0.37$). In contrast, Gradient AI showed notable regression. This underscores our hierarchical approach's ability to support long-context learning while maintaining general task performance.

## 4.4 ABLATION STUDIES

Our 100K context length single-document ablation studies, detailed in Appendix E, demonstrate that hierarchical ordering significantly boosts performance, particularly when combined with diverse question sets. Configurations with hierarchical ordering consistently outperformed those without, highlighting its importance for structuring instruction-tuning data. These findings provide a solid foundation for extending our experiments to larger context lengths and exploring the interaction of hierarchical and diverse question compositions. Building on these results, we expanded our experimentation to a 180K context length combining two documents, aiming to determine whether the patterns observed at 100K scale effectively with rope scaling. We also explore which question types (hierarchical or diverse and complex) perform best for questions directly following documents or referencing previous ones.

For each experiment, we generated 300–600 training samples of 180K tokens (concatenating two documents) using Qwen-2-72B and fine-tuned the data on LLaMA-3.1-8B-Instruct with a learning rate of 6e-5 for 1 epoch. As the 180K context length exceeds LLaMA-3.1-8B-Instruct's native 128K window, we applied rope scaling. The following compositions were tested: a) **Random vs. fixed number of questions**: Follow-up questions were either randomized (2–10) or fixed (6 main and 4 follow-up) to maintain consistency. b) **Hierarchical vs. diverse and complex questions**: We tested hierarchical ordering questions (h) against questions targeting specific, diverse, and complex reasoning (s). Each experiment is labeled as x-y-z, where x refers to questions following the first document, y the second, and z to questions referencing the first document after the second is processed. For instance, h-h-s-fixed includes 6 hierarchical questions for each document and 4 diverse follow-ups referencing the first document after the second. c) **Summarization**: Some experiments excluded global summarization at the start to assess its impact on model comprehension.

Table 7 shows the ablation results on InfiniteBench. Notably: 1) All experiments outperformed the baseline LLaMA-3.1-8B-Instruct model by a significant margin, demonstrating the effectiveness of our strategy with rope scaling. 2) Fixed questions outperform randomized ones: hs-hs-hs-fixed scored 59.45, surpassing hs-hs-hs-randomized (58.51). 3) Hierarchical questions paired with diverse questions achieve the best performance: hs-hs-hs-fixed yielded the highest score (59.45),

Table 7: Ablation study on InfiniteBench with 180K context length. Each experiment is labeled as x-y-z, where x is the type of question after the first document, y is the type of question after the second document, and z is the type of question referencing after the second document is processed. For example, h-h-s-fixed is the dataset with 6 hierarchical questions following the first document, 6 hierarchical questions following the second document, and 4 follow-up diverse questions referencing the first document after the second document is processed. Randomized signifies that the number of questions sampled is randomized, and no-sum signifies that the global summary is removed.

| Task | LLaMA-3.1-8B-Instruct | hs-hs-hs-randomized | hs-hs-hs-fixed | h-h-s-randomized |
|---|---|---|---|---|
| Retrieve.PassKey | **100.00** | **100.00** | **100.00** | **100.00** |
| Retrieve.Number | 95.33 | **100.00** | 99.33 | **100.00** |
| Retrieve.KV | 42.66 | 82.66 | **88.66** | 84.66 |
| En.Sum | **27.63** | 23.42 | 24.01 | 24.33 |
| En.QA | 24.83 | 33.32 | **34.26** | 31.84 |
| En.MC | 68.00 | 71.33 | **74.00** | 73.33 |
| En.Dia | 16.66 | 18.00 | 18.00 | 14.00 |
| Math.Find | 35.33 | **39.33** | 37.33 | 36.66 |
| Average | 51.31 | 58.51 | **59.45** | 58.10 |

| Task | h-h-s-fixed-no-sum | h-h-s-fixed | h-h-randomized | h-h-h-randomized |
|---|---|---|---|---|
| Retrieve.PassKey | **100.00** | **100.00** | **100.00** | **100.00** |
| Retrieve.Number | 99.33 | 99.33 | 98.66 | 99.33 |
| Retrieve.KV | 84.00 | 83.33 | 76.66 | 84.66 |
| En.Sum | 24.11 | 24.74 | 24.33 | 23.86 |
| En.QA | 32.81 | 33.88 | 30.69 | 31.97 |
| En.MC | 70.66 | 73.33 | 72.00 | 72.00 |
| En.Dia | 16.66 | 14.66 | 15.33 | **18.00** |
| Math.Find | 36.66 | **39.33** | 35.33 | 35.33 |
| Average | 58.03 | 58.58 | 56.63 | 58.14 |

highlighting the benefits of structuring and diverse, complex questions. 4) Summarization improves performance: hs-hs-fixed-no-sum scored 58.03, slightly below hs-hs-hs-fixed (58.58). Based on these findings, for longer context lengths (Section 4.2, we retain summarization, fix the number of questions/answers, and ensure both hierarchical and diverse questions are generated after direct documents and for those referencing previous ones.

## 5 CONCLUSION

This paper presents a novel strategy to generate high-quality, long-context instruction-tuning datasets that exceed the typical raw data context length. It incorporates hierarchical ordering to ensure logical coherence while maintaining diversity and complexity in questions. Systematic ablation studies show that combining diverse questions with hierarchical ordering enhances performance, particularly in long-context scenarios. Our 1M model demonstrates strong capabilities, outperforming LLaMA-3.1-8B-Instruct on InfiniteBench and significantly surpassing it on RULER, while maintaining robust performance on shorter-context tasks, as shown by LongBench and MMLU. Our data curation strategy is highly scalable, enabling efficient creation of instruction-tuning datasets exceeding 1 million tokens and scaling up to 10 million or more. With sufficient resources and a strong training stack, our method supports increasingly longer context lengths, potentially unlimited.

While our approach has significantly improved instruction tuning for long-context scenarios, a promising direction for future work is developing a self-evolutionary strategy that diversifies and adapts prompts. A short-context model could autonomously generate long-context instruction data using our methodology and evolve independently, creating diverse and adaptable prompts for various scenarios. This could enable models to progressively evolve into longer-context models. Additionally, combining our data-centric approach with architectural optimizations offers another promising avenue for future research.

**Ethics Statement.** In conducting this research, we ensured adherence to the highest ethical standards in the development and testing of our models. No human subjects were involved in data collection, ensuring that there are no privacy concerns or risks associated with the handling of personal information.

**Reproducibility.** We included the code to generate a bunch of hierarchical questions and diverse questions for a single document (see Section 3.1) in supplementary material (see generating-data.py). We also included the code to concatenate multiple documents (see Section 3.2) in supplementary material (see concatenate-350K.py). To enable long context training, we described detailed hardware setup in Section 4.1. Details about evaluations are also mentioned in in Section 4.1.

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

# A   APPENDIX: ADDITIONAL DETAILS ON DATA GENERATION ALGORITHMS

In this section, we present the pseudocode for the hierarchical QA generation strategy described in Section 3.1, along with the algorithm for combining multiple documents, as outlined in Section 3.2.

---

**Algorithm 1** Hierarchical Question Generation Strategy (Single Document)

---

```
 1: procedure GENERATEEXTENDEDCONTEXT(document, N_Questions_To_Generate)
 2:     chunks ← HierarchicalSplit(document.text)
 3:     summaries, full_summary ← SummarizeHierarchical(chunks)
 4:     conversations ← [InitialSummary(document.text, full_summary)]
 5:     for i = 1 to N_Questions_To_Generate do
 6:         context, summary ← SelectContext(chunks, summaries, last_medium, last_small, i)
 7:         qa_pair ← GenerateQAPair(context, summary)
 8:         AppendToConversations(conversations, qa_pair)
 9:         UpdateLastChunks(last_medium, last_small)
10:     end for
11:     return conversations
12: end procedure
13: procedure SELECTCONTEXT(chunks, summaries, last_medium, last_small, iteration_index)
14:     if first iteration then
15:         return random medium chunk
16:     else if no small chunk selected then
17:         return first small chunk of current medium
18:     else
19:         random_choice ← RandomChoice([0, 1, 2])          ▷ Equal 1/3 probability for each
20:         if random_choice = 0 then
21:             return deeper content of current small chunk
22:         else if random_choice = 1 then
23:             return next small chunk in current medium
24:         else
25:             return next medium chunk
26:         end if
27:     end if
28: end procedure
29: procedure GENERATEQAPAIR(context, summary)
30:     if ContextIsSpecific(context) then
31:         return GenerateSpecificQAPair(context)
32:     else
33:         return GenerateGeneralQAPair(context, summary)
34:     end if
35: end procedure
```

---

# B   ADDITIONAL INFORMATION ON DATA GENERATION PROMPTS

Here we list all prompts used in the different stages of our synthetic data generation pipeline.

> **Document Summarization**
>
> ```
> """Summarize the following text concisely in no
>     more than {word_limit} words:
>
>     {chunk}"""
> ```

---

**Algorithm 2** Concatenating Multiple Documents

---

**Input:** Set of $K$ documents, each with hierarchical and diverse questions
**Initialize:** conversation list $C \leftarrow \emptyset$
**for** each document $D_i$ where $i = 1, 2, \ldots, K$ **do**
    $H_i \leftarrow \text{GenerateHierarchicalQuestions}(D_i)$
    $S_i \leftarrow \text{RandomlySampleSpecificQuestions}(D_i)$
    $C \leftarrow C \cup \text{InitialHierarchicalQuestions}(H_i)$
    $C \leftarrow C \cup \text{RandomlySampleDiverseQuestions}(S_i)$
    Store remaining unselected diverse questions from $S_i$
**end for**
**for** each document $D_i$ where $i = 2, 3, \ldots, K$ **do**
    $C \leftarrow C \cup \text{NextHierarchicalQuestions}(H_{i-1})$
    $C \leftarrow C \cup \text{RandomlySampleUnselectedDiverse}(S_{i-1})$
    Update hierarchical index for document $D_i$
**end for**
**for** each document $D_i$ where $i = 1, 2, \ldots, K - 1$ **do**
    **if** RandomCondition(0.6) **then**
        $C \leftarrow C \cup \text{FollowUpHierarchicalQuestions}(H_i)$
    **end if**
**end for**
Process remaining specific and diverse questions:
$x \leftarrow \frac{\text{Length}(S_i)}{2}$
**if** $x \geq \text{ThresholdForSpecificQuestions}$ **then**
    Select and append follow-up specific questions to $C$
    Remove selected follow-up specific questions from pool
**end if**
**Output:** Final conversation list $C$

---

**Diverse Questions**

```
"""Context information is below.
--------------------
${context}
--------------------
Given the context information and not prior knowledge.
Generate content based on the below query.
You are a Teacher/Professor. Your task is to
set up 1 diverse temporal question about the
context for an upcoming quiz/examination. The question
should cover different time periods and events
described in the context. Restrict the question
to the context information provided. You must
return the result in JSON: {'question': <question>,
'answer': <answer>}"""
```

---

**Diverse Questions**

```
"""Context information is below.
---------------------
${context}
---------------------
Given the context information and not prior knowledge.
Generate content based on the below query.
You are a Teacher/Professor. Your task is to
create 1 character-based question from the context
for an upcoming quiz/examination. The question should
explore different aspects of the characters, such
as their motivations, actions, and relationships. Restrict
the question to the context information provided.
You must return the result in JSON:
{'question': <question>, 'answer': <answer>}"""

"""Context information is below.
---------------------
${context}
---------------------
Given the context information and not prior knowledge.
Generate content based on the below query.
Formulate 1 complex question that requires analysis
of multiple aspects from the context for
an upcoming quiz/examination. The question should encourage
critical thinking and synthesis of different pieces
of information within the context. Restrict the
question to the context information provided. You
must return the result in JSON: {'question':
<question>, 'answer': <answer>}"""

"""Context information is below.
---------------------
${context}
---------------------
Given the context information and not prior knowledge.
Generate content based on the below query.
You are a Teacher/Professor. Ask 1 question
about the main themes or messages of
the text for an upcoming quiz/examination. The
question should cover different aspects of the
themes and how they are developed in
the context. Restrict the question to the
context information provided. You must return the
result in JSON: {'question': <question>,
'answer': <answer>}"""
```

---

> **Diverse Questions**
>
> ```
> """Context information is below.
> ---------------------
> ${context}
> ---------------------
> Given the context information and not prior knowledge.
> Generate content based on the below query.
> You are a Teacher/Professor. Create 1 question
> that compare different elements within the context
> for an upcoming quiz/examination. The question should
> highlight similarities and differences between various
> elements such as characters, events, and themes. Restrict
> the question to the context information provided.
> You must return the result in JSON:
> {'question': <question>, 'answer': <answer>}"""
>
> """Context information is below.
> ---------------------
> ${context}
> ---------------------
> Given the context information and not prior knowledge.
> Generate content based on the below query.
> You are a Teacher/Professor. Develop 1 question
> that explore the cause and effect relationships
> within the context for an upcoming quiz/examination.
> The question should focus on understanding the
> reasons behind events and their outcomes. Restrict
> the question to the context information provided.
> You must return the result in JSON:
> {'question': <question>, 'answer': <answer>}"""
>
> """Context information is below.
> ---------------------
> ${context}
> ---------------------
> Given the context information and not prior knowledge.
> Generate content based on the below query.
> You are a Teacher/Professor. Create 1 hypothetical
> question based on the context for an
> upcoming quiz/examination. The question should explore
> what-if scenarios and possible alternate outcomes. Restrict
> the question to the context information provided. You
> must return the result in JSON: {'question':
> <question>, 'answer': <answer>}"""
> ```

**Diverse Questions**

```
"""Context information is below.
---------------------
${context}
---------------------
Given the context information and not prior knowledge.
Generate content based on the below query.
You are a Teacher/Professor. Formulate 1 question
that require interpretation of the context for
an upcoming quiz/examination. The question should encourage
students to provide their own insights and
interpretations based on the information given. Restrict
the question to the context information provided.
You must return the result in JSON:
{'question': <question>, 'answer': <answer>}"""

"""Context information is below.
---------------------
${context}
---------------------
Given the context information and not prior knowledge.
Generate content based on the below query.
You are a Teacher/Professor. Ask 1 detail-oriented
question about the context for an upcoming
quiz/examination. These question should focus on specific
details, facts, and figures mentioned in the
context. Restrict the question to the context
information provided. You must return the result
in JSON: {'question': <question>, 'answer': <answer>}"""

"""Context information is below.
---------------------
${context}
---------------------
Given the context information and not prior knowledge.
Generate content based on the below query.
You are a Teacher/Professor. Create 1 question
that explore different perspectives or viewpoints within
the context for an upcoming quiz/examination. The
question should examine how different characters or
groups might view events or themes differently.
Restrict the questions to the context information
provided. You must return the result in
JSON: {'question': <question>, 'answer': <answer>}"""
```

**Multi-Hop Questions**

```
"""Context information is below.
${selected_chunk_1}
${selected_chunk_2}
${selected_chunk_3}
You are a Professor designing a final exam
for an advanced interdisciplinary course. Create 1
complex question requiring deep analysis and synthesis of
information from all three chunks. Do not mention
that there are three chunks/your questions. Do not
mention excerpts either. For example, instead of a
question that says "Analyze the theme of justice
and its various forms as portrayed in the
three provided literary excerpts. How do the characters'
actions and the outcomes of their situations reflect
or challenge traditional notions of justice? Consider the
legal, personal, and societal implications of justice in
each excerpt and discuss the role of power
dynamics in shaping justice." You should say: "Analyze
the theme of justice and its various forms
as portrayed. How do the characters' actions and
the outcomes of their situations reflect or challenge
traditional notions of justice? Consider the legal, personal,
and societal implications of justice and discuss the
role of power dynamics in shaping justice."
Question Guidelines:
1. The question must integrate and require reasoning
across all three chunks.
2. The question should be multi-layered, promoting analysis,
synthesis, and evaluation.
Answer Guidelines:
1. Provide a comprehensive answer addressing all question
aspects.
2. Reference and interconnect information from each chunk.
Return 1 question-answer pair in JSON format:
{ "question": <question>, "answer": <answer> }
"""
```

---

**Specific Detail Question**

```
"""Context information is below.
${context}
Given the context information and not prior knowledge,
generate content based on the below query. You
are a Teacher/Professor. Create 1 specific question about
the details, events, characters, and settings from the
context provided. This question should have an exact,
unambiguous answer that can be directly found in
the given information. The question should be similar
in style to the following examples:
"Where does Character A meet Character B for
the first time?"
"What is Character C's religion?"
"Where does Character D live for the majority
of the story?"
"Which of the following is NOT one of
Character E's responsibilities?"
"Which among [list of names] is not [Character
F]'s child?"
"Who among [list of names] is the final
to perish?"
"What's the name of [Family name]'s summer home?"
"Who accompanied [Character G] [specific activity] at last?"
Ensure that the question and answer are strictly
based on the context information provided. The question
may include multiple-choice options when appropriate. You
must return the result in JSON: {'question': <question>,
'answer': <answer>}"""
```

## C COMPLETE CONVERSATION DATA

Here we present an example of our instruction tuning data that is developed by concatenating four documents (350K context length). The data consists of three types of content: long documents and their summarizations (yellow) (constituting QA pairs), QA pairs extracted from each document (blue), and QA pairs that reference past documents (green). These are color-coded in the following example for clarity.

```
{"conversations": [
  {
    "role": "user",
    "content": "Produced by Charles Bowen from page scans provided by the
        Web Archive\n\n\n\n\nTranscriber's Notes: ... Please give me a
        summary of the book"
  },
  {
    "role": "assistant",
    "content": "Charles Bowen created a digital book from Web Archive
        scans, ..."
  },
```

```
  {
    "role": "user",
    "content": "What are the main reasons for the tensions and conflicts
        among the characters in 'The Patricians'?"
  },
  {
    "role": "assistant",
```

```
    "content": "The tensions and conflicts in 'The Patricians' arise from
        issues of pride, ..."
  },
  {
    "role": "user",
    "content": "What class in ancient Rome did the term 'Patricians'
        typically refer to?"
  },
  {
    "role": "assistant",
    "content": "The term 'Patricians' typically referred to the
        aristocratic class in ancient Rome, who were affluent, powerful,
        and held considerable political sway in the Roman Republic."
  },
  {
    "role": "user",
    "content": "Who among the following did not attend Christopher's
        banquet? A) Tausdorf B) Althea C) Schindel D) Francis"
  },
  {
    "role": "assistant",
    "content": "D) Francis"
  },
  {
    "role": "user",
    "content": "Analyze the theme of justice and its various forms as
        portrayed. How do the characters' actions and the outcomes of
        their situations reflect or challenge traditional notions of
        justice? Consider the legal, personal, and societal implications
        of justice and discuss the role of power dynamics in shaping
        justice."
  },
  {
    "role": "assistant",
    "content": "The theme of justice is explored through the complex
        interplay of legal, personal, and societal implications,
        reflecting and challenging traditional notions of justice. ..."
  },
  ...
```

```
  {
    "role": "user",
    "content": "Produced by D Alexander and the Online Distributed
        Proofreading Team at http://www.pgdp.net ... + Please give me a
        summary of the book"
  },
  {
    "role": "assistant",
    "content": "The text acknowledges D Alexander ..."
  },
```

```
  {
    "role": "user",
    "content": "Who does the professor and his party meet at the sulphur
        spring?"
  },
  {
    "role": "assistant",
    "content": "Jranvin, the chief of the islanders"
  },
  {
    "role": "user",
```

```
  "content": "What does Jim recognize that lifts his spirits while
      confined
  on the Marjorie?"
},
{
  "role": "assistant",
  "content": "Jim recognizes his own ship, which lifts his spirits
      while he is
  confined on the Marjorie."
},
{
  "role": "user",
  "content": "How did the professor and his party find out about the
      location
  of the Marjorie and what did they discover upon investigating the
      harbor?"
},
{
  "role": "assistant",
  "content": "The professor and his party, including Tom, Jo, and
      Berwick,
  hiked to a cliff and climbed to the top where they spotted the
      Marjorie
  and another ship in a harbor five miles away. Upon investigating the
      harbor,
  they found the Sea Eagle leaving and chaos on the Marjorie. They also
       captured
  the steward who revealed he helped Jim escape."
},
...
```

```
{
  "role": "user",
  "content": "Analyze the theme of justice and its various forms as
      portrayed.
  How do the characters' actions and the outcomes of their situations
      reflect
  or challenge traditional notions of justice? Consider the legal,
      personal,
  and societal implications of justice and discuss the role of power
      dynamics
  in shaping justice."
},
{
  "role": "assistant",
  "content": "The theme of justice is intricately woven through
  the narrative, ..."
},
...
```

```
{
  "role": "user",
  "content": "\n\n\n\nProduced by Marius Masi, Juliet Sutherland and
      the Online
  \nDistributed Proofreading Team at ...  Please give me a summary of
      the book"
},
{
  "role": "assistant",
  "content": "The text describes a collection of illustrated books
      about Japan, ..."
},
```

```
{
  "role": "user",
  "content": "How does the text compare Japanese and Western art in
      terms
  of their integration into daily life and appreciation by the public?"
},
{
  "role": "assistant",
  "content": "The text emphasizes that Japanese art, unlike Western art
      ,
  is deeply integrated into daily life and universally appreciated,
  similar to the vitality of art in ancient Greece..."
},
{
  "role": "user",
  "content": "How does the book contrast Japanese and Western art in
      terms of
  their integration into daily life and societal appreciation?"
},
{
  "role": "assistant",
  "content": "The book highlights that Japanese art is more integrated
      into daily
  life and universally appreciated, ..."
},
...
```

```
{
  "role": "user",
  "content": "Analyze the character development of Tausdorf throughout
      the
  passage. How does his perception of Bona change, and what factors
      contribute
  to this change? Additionally, discuss the role of the setting and the
       use of
  imagery in shaping Tausdorf's actions and emotions."
},
{
  "role": "assistant",
  "content": "Tausdorf's character development is complex and
      multifaceted, ..."
},
{
  "role": "user",
  "content": "What is the name of the yacht that Jim and Berwick
      consider hiring?"
},
{
  "role": "assistant",
  "content": "Storm King"
},
...
```

```
{"role": "user", "content": "\n\n\n\nProduced by David Edwards and the
     Online Distributed ... Please give me a summary of the book"},
{"role": "assistant", "content": "\"The Curlytops Snowed In\" by Howard
     R. Garis follows siblings Ted and Janet Martin, known as the
     Curlytops, ..."},
```

```
{"role": "user", "content": "What challenge did Nicknack, the goat,
     face during the winter adventures, and how did the children address
      it?"},
```

| 1M model | | 650K model | | 350K model | |
|---|---|---|---|---|---|
| Context Length | Accuracy (%) | Context Length | Accuracy (%) | Context Length | Accuracy (%) |
| 1 000 000 | 62.95% | 650 000 | 57.24 % | 350 000 | 71.15 % |
| 524 288 | 76.88% | 524 288 | 74.88 % | 262 144 | 78.26 % |
| 262 144 | 76.24 % | 262 144 | 78.26 % | 131 072 | 82.72 % |
| 131 072 | 82.05 % | 131 072 | 82.29 % | 65 536 | 82.63 % |
| 65 536 | 83.73 % | 65 536 | 81.81 % | 32 768 | 87.26 % |
| 32 768 | 86.19 % | 32 768 | 87.96 % | 16 384 | 91.45 % |
| 16 384 | 90.06 % | 16 384 | 92.86 % | 8 192 | 92.62 % |
| 8 192 | 92.96 % | 8 192 | 91.92 % | 4 096 | 96.13 % |
| 4 096 | 95.59 % | 4 096 | 95.49 % | | |

Table 8: Context length accuracy of RULER for different models

```
{"role": "assistant", "content": "Nicknack had difficulty standing on
    ice due to the unsuitability of his hooves..."},
{"role": "user", "content": "What causes the snow house to collapse in
    Chapter VI, and what do the children decide to do after the
    incident?"},
{"role": "assistant", "content": "The snow house collapses when
    Nicknack the goat ..."},
...
```

```
{"role": "user", "content": "Compare and contrast the portrayal of art
    and its integration into daily life in Japan and the West, as
    described in the text. How does the Japanese approach to art,
    particularly in the context of theater, reflect broader cultural
    values and philosophies? Discuss the role of realism, balance, and
    the environment in shaping the artistic experience in both cultures
    ."},
{"role": "assistant", "content": "The text illustrates a stark contrast
     between the Japanese and Western approaches to art, particularly
    in theater, which reflects deeper cultural values and philosophies.
     In Japan, art is not merely a conscious effort to achieve beauty
    but an instinctive expression of inherited taste that permeates all
     aspects of life..."},
{"role": "user", "content": "Analyze the theme of survival and its
    various forms as portrayed in the narratives. How do the characters
    ' actions and the outcomes of their situations reflect or challenge
     traditional notions of survival? Consider the natural, social, and
     personal implications of survival and discuss the role of
    resourcefulness in shaping survival strategies."},
{"role": "assistant", "content": "The theme of survival is intricately
    woven through the narratives, reflecting the characters' resilience
     and adaptability in the ..."},
 ...
]}
```

# D  RULER NUMERICAL RESULTS

# E  100K CONTEXT LENGTH ABLATION STUDIES

The 100K ablation studies aim to assess whether hierarchical ordering and diverse question types improve results on single-document instruction tuning data. We also aim to identify which of these factors most significantly influences overall performance. In particular, we want to explore (1) whether hierarchical ordering enhances outcomes, (2) whether diverse question sets contribute positively, and (3) whether the use of multi-hop questions further boosts results.

Table 9: Context length of RULER for LLaMA-3.1-8B-Instruct models

| LLaMA-3.1-8B-Instruct | |
|---|---|
| Context Length | Percentage (%) |
| 524 288 | 28.93 % |
| 262 144 | 30.34 % |
| 131 072 | 83.06 % |
| 65 536 | 82.26 % |
| 32 768 | 88.44 % |
| 16 384 | 93.13 % |
| 8 192 | 92.08 % |
| 4 096 | 95.49 % |

| Zero-shot Rope Scaling to 1M | |
|---|---|
| Context Length | Percentage (%) |
| 1 000 000 | 48.81 % |
| 524 288 | 62.53 % |
| 262 144 | 66.44 % |
| 131 072 | 72.68 % |
| 65 536 | 77.81 % |
| 32 768 | 84.01 % |
| 16 384 | 87.36 % |
| 8 192 | 90.73 % |
| 4 096 | 95.94 % |

(a) Context length of RULER of LLaMA-3.1-8B-Instruct

(b) Context length of RULER with zero-shot rope scaling to 1M context length

Each experiment uses 300-600 data samples, each with 100K tokens, fine-tuned on LLaMA-3.1-8B-Instruct for 1 epoch at a 6e-5 learning rate. The specific ablation tests we included are 1) **4 hierarchies**: from a single document, we generated hierarchical ordering data using the algorithm specified in Section 3.1. 2) **4 hierarchies with multi-hop reasoning**: In addition to the 4 hierachies set up in Section 3.1, every time we generate a new QA pair, there is a 20 % chance that a multi-hop question-answer pair will follow. 3) **4 hierarchies without order**: hierarchical questions were generated without enforcing the order from Section 3.1, testing if strict hierarchy enforcement improves outcomes. 4) **Diverse questions**: this setup generated various question types to test if diversity improves performance, as outlined in Section 3.1.

The results of these ablation studies on InfiniteBench are summarized in Table 11. The key findings include: 1) Multi-Hop Reasoning Improves Performance: Among all configurations, multi-hop reasoning achieved the highest average score of 54.70, demonstrating the importance of capturing cross-document relationships and broader reasoning capabilities. 2) Diverse Questions Provide Broad Improvements: The diverse questions setup achieved the second-highest score of 52.41, highlighting the value of introducing variety in QA generation for instruction-tuning data. 3) Hierarchical Ordering Boosts Performance: Both the strict hierarchical model (52.08) and the random hierarchical model (50.69) outperformed the base LLaMA-3.1-8B-Instruct (51.31), validating the effectiveness of hierarchical structuring, even when not strictly ordered.

The LongBench results (presented in Table 10) provide additional insights, though the differences between configurations are relatively minor. This is likely because LongBench evaluates models on short contexts (up to 10K tokens), which do not fully leverage the strengths of hierarchical or multi-hop structures designed for longer contexts. In summary, the ablation tests show that hierarchical ordering, multi-hop reasoning, and diverse questions are key to optimizing performance on long-context tasks.

Table 10: Ablation study on LongBench with 100K context length.

| Task | LLaMA-3.1-8B-Instruct | 4 hierarchies | 4 hierarchies multi-hop | 4 hierarchies random | diverse questions |
|---|---|---|---|---|---|
| NarrativeQA | 25.48 | 25.89 | 25.10 | 25.04 | **27.91** |
| Qasper | 45.33 | **47.02** | 44.79 | 46.00 | 46.25 |
| MultiFieldQA-en | **54.98** | 54.86 | 53.96 | 54.86 | 53.75 |
| MultiFieldQA-zh | **61.83** | 55.75 | 54.87 | 59.89 | 56.14 |
| Single Document | **46.91** | 45.88 | 44.68 | 46.45 | 46.01 |
| HotpotQA | 55.00 | 56.67 | 56.91 | 55.83 | **58.34** |
| 2WikiMQA | 44.95 | 52.19 | **52.96** | 48.74 | 52.71 |
| Musique | **31.76** | 29.15 | 28.55 | 29.85 | 28.10 |
| DuReader | 34.10 | **36.83** | 36.32 | 35.57 | 36.74 |
| Multi-Document | 41.45 | 43.71 | 43.69 | 42.50 | **43.97** |
| GovReport | 35.07 | 34.39 | 33.72 | 35.31 | **35.33** |
| QMSum | 25.13 | 25.15 | 25.27 | **25.52** | 25.38 |
| MultiNews | 27.08 | 27.34 | **27.48** | 27.29 | 27.46 |
| VCSUM | **17.10** | 16.12 | 16.75 | 16.13 | 16.40 |
| Summarization | 26.10 | 25.75 | 25.81 | 26.06 | **26.14** |
| TREC | 72.50 | **73.00** | **73.00** | **73.00** | 72.00 |
| TriviaQA | 91.65 | **92.28** | 92.25 | 91.87 | 91.83 |
| SAMSum | 43.77 | 43.81 | 43.98 | 44.49 | **45.48** |
| LSHT | 46.00 | 46.00 | 47.00 | 47.00 | **48.00** |
| Few-shot Learning | 63.48 | 63.77 | 64.06 | 64.09 | **64.33** |
| Passage Count | 6.55 | 4.00 | 3.00 | **7.56** | 5.00 |
| PassageRetrieval-e | **99.50** | 99.00 | 99.00 | 98.50 | 98.50 |
| PassageRetrieval-z | 96.38 | 98.50 | **100.00** | 94.63 | 99.50 |
| Synthetic Tasks | 67.48 | 67.17 | 67.33 | 66.90 | **67.67** |
| All | 48.11 | 48.31 | 48.15 | 48.27 | **48.67** |

Table 11: Ablation study on InfiniteBench with 100K context length.

| | LLaMA-3.1-8B-Instruct | 4 hierarchies | diverse questions | 4 hierarchies random | 4 hierarchies multi-hop |
|---|---|---|---|---|---|
| Retrieve.PassKey | **100.00** | 86.66 | 86.66 | 86.66 | **100.00** |
| Retrieve.Number | 95.33 | 86.66 | 86.00 | 85.33 | **96.66** |
| Retrieve.KV | 42.66 | **60.00** | 58.00 | 58.66 | 57.33 |
| En.Sum | **27.63** | 23.02 | 24.11 | 22.77 | 22.67 |
| En.QA | 24.83 | 29.66 | **32.50** | 25.40 | 30.25 |
| En.MC | 68.00 | 70.66 | **72.00** | 70.00 | 70.66 |
| En.Dia | 16.66 | 24.66 | 23.33 | 20.66 | **26.00** |
| Math.Find | 35.33 | 35.33 | **36.66** | 36.00 | 34.00 |
| Average | 51.31 | 52.08 | 52.41 | 50.69 | **54.70** |

