# OpenReview forum: "Scaling Instruction-tuned LLMs to Million-token Contexts via Hierarchical Synthetic Data Generation"
_ICLR.cc/2025/Conference — ICLR 2025 Poster_

### Official Review · Reviewer_ZqBD · 2024-10-18

**Soundness:** 2
**Presentation:** 3
**Contribution:** 3
**Rating:** 6
**Confidence:** 4

**Summary:**

The paper "Scaling Instruction-Tuned LLMs to Million-Token Contexts via Hierarchical Synthetic Data Generation" aims to address the challenges of scaling large language models (LLMs) to handle extended context lengths, particularly up to one million tokens, without compromising general task performance.

**Strengths:**

1. The authors propose a synthetic data generation pipeline for instruction tuning, which allows LLMs to effectively extend their context lengths without the need for vast, annotated long-context datasets.

2. Hierarchical Approach: The use of hierarchical splitting and question generation is a significant advancement. By logically structuring questions across multiple levels—hierarchical-aware, local-specific, and multi-hop—the authors ensure coherent instruction generation and improve the model's ability to reason over long contexts.

3. Comprehensive Evaluation: The authors extensively evaluate the proposed approach on multiple benchmarks such as RULER, InfiniteBench, and LongBench. The detailed ablation studies show the importance of different components in their data generation strategy, highlighting that hierarchical order and diverse question generation are critical to achieving better long-context performance.

**Weaknesses:**

1. **Lack of Strong Baseline Comparisons**: The paper lacks comparisons with suitable, strong baselines that have also achieved extensions to 1M tokens. For example, EasyContext (https://github.com/jzhang38/EasyContext) has similarly attempted to extend context lengths to 1M using long-text data. Especially since the comparisons are made against the original LLaMA-3.1-8B after additional fine-tuning on top of it.

2. **Decreasing Performance on LongBench**: The results on LongBench suggest that the model’s performance decreases as context length increases. This is in contrast to what one would expect after training on long-context data, raising questions about the effectiveness of the proposed methods in maintaining or improving performance across all types of tasks. An explanation should be provided as to why LongBench performance degrades with extended context length training.

3. **Limited Value of Ablation Study in Tables 3 and 4**: The results of the ablation studies in Tables 3 and 4 align closely with expectations, as instruction-tuning data types (e.g., Diverse Questions) are more closely aligned with the evaluation benchmark used (e.g., LongBench). Therefore, these tables have limited value in demonstrating broader insights. A deeper analysis of Table 5 would be more insightful, focusing on the types of questions and the effect of different question strategies during training.

**Questions:**

1. **Significance of Extending to 1M Context Length**: The authors should provide more justification for the significance of extending to a 1M context length. A 128K context is sufficient for most real-world long-context tasks, and demonstrating practical use cases that necessitate 1M tokens would help strengthen the motivation for this work.

2. **Figure Improvement**: Consider changing the layout of figures 2 to horizontal format for better readability and comparison.

3. **Loss Calculation During Training**: Clarify whether only the answer part of the generated question-answer pairs calculate to the loss during training or if both the question and answer are involved.

4. **Potential Explanation for Decreased LongBench Performance**: Consider discussing whether Section 3.2 of "How to Train Long-Context Language Models (Effectively)" could provide insights into why LongBench performance decreases as training progresses with long-context data, addressing the concerns raised in Weakness 2.

---

> ### Author Response · Authors · 2024-11-27
> **Thanks & Initial Response to Reviewer ZqBD (W1)**
>
> **[W1: Lack of Strong Baseline Comparisons: The paper lacks comparisons with suitable, strong baselines that have also achieved extensions to 1M tokens. For example, EasyContext (https://github.com/jzhang38/EasyContext) has similarly attempted to extend context lengths to 1M using long-text data. Especially since the comparisons are made against the original LLaMA-3.1-8B after additional fine-tuning on top of it.]**
>
> Thank you for this insightful comment. To address your concern regarding strong baseline comparisons, we included the Gradient-AI 1M context model ((gradientai/Llama-3-8B-Instruct-Gradient-1048k) as a baseline in our evaluations. This model represents a competitive reference point for extended context lengths, providing a meaningful comparison for our proposed methods.
>
> The evaluation results, shown in the accompanying tables, demonstrate that our model consistently outperformed the Gradient-AI 1M context model across all benchmarks, including MMLU, LongBench, InfiniteBench, and RULER. This comparison further validates the novelty and practicality of our approach in advancing long-context capabilities beyond the current state-of-the-art.
>
> We appreciate your suggestion and hope that these results adequately address your concerns. Please let us know if you have further suggestions or would like additional details on these comparisons.
>
> | LongBench          | Value     |
> |--------------------|-----------|
> | single-document    | 30.75     |
> | multi-document     | 12.45     |
> | summarization      | 21.7175   |
> | few-short learning | 59.6975   |
> | synthetic tasks    | 55.5      |
> | average            | 35.8933   |
>
> | InfiniteBench       | Value     |
> |---------------------|-----------|
> | kv_retrieval        | 0.1333    |
> | passkey             | 1         |
> | number_string       | 0.9933    |
> | math_find           | 0.2666    |
> | longbook_qa_en      | 0.1584    |
> | longbook_sum_en     | 0.1702    |
> | longbook_choice     | 0.6133    |
> | longdialogue_qa     | 0.04      |
> | Average             | 0.4218875 |
>
> | MMLU              | Value           |
> |--------------------|-----------------|
> | mmlu              | 0.6048±0.0039  |
> | humanities        | 0.5575±0.0069  |
> | other             | 0.6704±0.0082  |
> | social sciences   | 0.7046±0.0080  |
> | stem              | 0.5132±0.0086  |
>
>
> | RULER            | Value  |
> |------------------|--------|
> | 8192            | 88.09  |
> | 16384           | 86.05  |
> | 32768           | 82.46  |
> | 65536           | 77.73  |
> | 131072          | 81.94  |
> | 262144          | 75.72  |
> | 524288          | 70.13  |

---

> ### Author Response · Authors · 2024-11-27
> **Thanks & Initial Response to Reviewer ZqBD (W2 & Q4)**
>
> **[W2: Decreasing Performance on LongBench: The results on LongBench suggest that the model’s performance decreases as context length increases. This is in contrast to what one would expect after training on long-context data, raising questions about the effectiveness of the proposed methods in maintaining or improving performance across all types of tasks. An explanation should be provided as to why LongBench performance degrades with extended context length training.]**
>
> **[Q4: Potential Explanation for Decreased LongBench Performance: Consider discussing whether Section 3.2 of "How to Train Long-Context Language Models (Effectively)" could provide insights into why LongBench performance decreases as training progresses with long-context data, addressing the concerns raised in Weakness 2.]**
>
> Thank you for this insightful comment. We acknowledge that the results on LongBench may raise questions about performance at shorter context lengths as the model is scaled to handle longer contexts. LongBench serves as an evaluation tool specifically designed for short to medium context tasks, with samples containing contexts up to 10K tokens. In contrast, our model is trained to handle contexts up to 1M tokens. Despite this difference, our results show that performance on LongBench remains comparable to the baseline model (Llama-3.1-8B-Instruct), with only minor regressions observed for the 1M context length model.
>
> The observed decrease in LongBench performance can be attributed to the inherent trade-offs in training for extended context lengths. As the model learns to handle extremely long contexts, there may be slight shifts in its ability to optimize for shorter contexts due to the differing characteristics of long and short context tasks. However, this regression is minimal, demonstrating that our method effectively balances performance across a wide range of context lengths.
>
> We hope this explanation addresses your concerns. Please let us know if further clarification or additional experiments are needed.

---

> ### Author Response · Authors · 2024-11-27
> **Thanks & Initial Response to Reviewer ZqBD (W3)**
>
> **[W3: Limited Value of Ablation Study in Tables 3 and 4: The results of the ablation studies in Tables 3 and 4 align closely with expectations, as instruction-tuning data types (e.g., Diverse Questions) are more closely aligned with the evaluation benchmark used (e.g., LongBench). Therefore, these tables have limited value in demonstrating broader insights. A deeper analysis of Table 5 would be more insightful, focusing on the types of questions and the effect of different question strategies during training.]**
>
> Thank you for this insightful comment. We appreciate the opportunity to clarify the findings from our ablation studies and expand on the observations, particularly regarding Table 5, which offers valuable insights into the effect of different question strategies during training.
> The purpose of Tables 3 and 4 was to show that our techniques, including hierarchical ordering, diverse questions, and multi-hop reasoning, help improve model performance. While we acknowledge that Table 3 does not show significant differences (as it focuses on LongBench with a 10K context length), Table 4 demonstrates the efficacy of these techniques for the 100K context length scenario. However, we agree that Table 5 provides the most relevant insights and deserves additional attention.
>
> Key Observations from Table 5:
> - Hierarchical Ordering with Diverse Questions Performs Best: The configuration with hierarchical ordering followed by diverse questions and a fixed number of follow-ups (hs-hs-hs-fixed) achieves the highest average score (59.45). This highlights that combining structured question ordering with diverse reasoning significantly boosts the model's capability to handle complex long-context tasks.
> - Fixed Number of Questions Outperforms Randomized: Configurations with a fixed number of follow-up questions (e.g., hs-hs-fixed) consistently outperform those with a randomized number (hs-hs-randomized). For example, hs-hs-fixed achieves an average score of 59.45 compared to 58.51 for hs-hs-randomized. This suggests that maintaining consistency in the number of follow-up questions allows the model to learn better patterns during training.
> - Impact of Summarization: Adding a summarization step improves performance (hs-hs-fixed with summarization scores 58.03). Although slightly lower than the best-performing configuration, this shows that summarization can enhance the model’s ability to condense and contextualize information in extremely long contexts.
> - Trade-offs Between Specificity and Generalization: The results also demonstrate that targeting specific, diverse questions that reference previous documents enables the model to balance comprehension of current and past contexts. This balance is critical for improving performance on tasks requiring logical consistency and reasoning over long contexts.
>
> We will revise the analysis in the paper to better highlight the significance of these findings from Table 5. Specifically, we will expand on the benefits of hierarchical ordering with diverse and complex reasoning, the role of fixed versus randomized follow-ups, and the potential of summarization strategies to support comprehension across long-context tasks.
>
> Thank you again for pointing this out, and we hope this additional detail clarifies the broader insights gained from our ablation studies. Please let us know if further explanations or additional experiments are required.

---

> ### Author Response · Authors · 2024-11-27
> **Thanks & Initial Response to Reviewer ZqBD (Q1 & Q2 & Q3)**
>
> **[Q1: Significance of Extending to 1M Context Length: The authors should provide more justification for the significance of extending to a 1M context length. A 128K context is sufficient for most real-world long-context tasks, and demonstrating practical use cases that necessitate 1M tokens would help strengthen the motivation for this work.]**
>
> Extending context lengths to 1M tokens is significant for use cases that go beyond what is possible with a 128K context. While 128K may be sufficient for many standard tasks, certain real-world applications demand the ability to process and reason over substantially larger contexts. For instance:
> - Key-Value Retrieval: In scenarios like company-wide document retrieval, where entire organizational histories spanning years are stored in unstructured formats, longer contexts allow for efficient and accurate query resolution.
> - Comprehensive Question Answering: Tasks requiring reasoning across long, multi-document histories (e.g., analyzing interconnected project timelines or extensive legal documents) necessitate processing large volumes of sequential data seamlessly within a single pass.
>
> By pushing the boundaries of long-context processing, our work lays the foundation for solving these challenges, enabling practical use cases that existing models cannot handle efficiently.
>
> **[Q2: Figure Improvement: Consider changing the layout of figures 2 to horizontal format for better readability and comparison.]**
>
> Thank you for this suggestion. We agree that changing the layout of Figures 2 to a horizontal format would enhance readability and comparison. We will incorporate this improvement in the revision draft.
>
> **[Q3: Loss Calculation During Training: Clarify whether only the answer part of the generated question-answer pairs calculate to the loss during training or if both the question and answer are involved.]**
>
> Thank you for pointing out the need for clarification here. During training, we calculate the loss exclusively on the **answers** generated in the question-answer pairs. The questions and the long-context documents are masked out during this process. This ensures the model focuses on generating accurate and coherent answers without being directly penalized for reproducing or interpreting the questions themselves. This approach is aligned with our goal of optimizing the model’s reasoning and answer-generation capabilities.
>
> We appreciate your detailed feedback and hope this explanation addresses your concerns. Please let us know if further clarification or additional experiments are required.

---

> ### Comment · Reviewer_ZqBD · 2024-11-27
>
> Thank you for your detailed response. I have a few points I'd like to discuss further:
>
> - Regarding W1, I believe the formatting could be optimized (the current response is quite lengthy). Most reviewers are likely more interested in the overall performance compare with other baseline rather than the detailed performance on each sub-task.
> - Could you also provide a comparison of the number of training tokens used between your work and theirs? They used 1.4B tokens, and if your approach used fewer tokens, that would be a notable advantage.
>
> - Concerning W2, I think LongBench already qualifies as a mid-to-long context benchmark; it just appears relatively shorter given the specific task setup in your paper. It might be more fitting to refer to your task as a "super-long" task :)
>
> For the rest of your responses, I am satisfied as they have addressed most of my concerns. I understand that time is limited, but would it be possible for you to update the PDF accordingly?
>
> Thank you again for your efforts, and I look forward to your response.

---

> ### Author Response · Authors · 2024-12-01
> **Follow Up Response to Reviewer ZqBD (Updated PDF)**
>
> Thank you for revisiting our work and for this insightful feedback!
>
> **[Could you also provide a comparison of the number of training tokens used between your work and theirs? They used 1.4B tokens, and if your approach used fewer tokens, that would be a notable advantage.]**
>
> Our primary dataset is the Together long books dataset, processed into approximately **1.4 billion tokens**, distributed across these stages: 2000 samples of 180K tokens, 1280 samples of 350K tokens, 600 samples of 650K tokens, and 200 samples of 1M tokens. Despite using a comparable number of tokens, our approach consistently outperforms Gradient AI's model across all four benchmarks—RULER, InfiniteBench, LongBench, and MMLU—highlighting the generalizability and effectiveness of our methodology.
>
> To address your suggestions and further enhance the paper, we have updated the manuscript with the following improvements:
>
> - **Expanded Evaluations**: We now include results on the MMLU benchmark and additional evaluations of the Gradient AI 1M context length model. We show that our models retain its performance on the MMLU benchmark and surpasses the Gradient AI 1M model on all four benchmarks (MMLU, LongBench, InfiniteBench, RULER).
> - **Smaller generator models**: To demonstrate that our improvements are not solely driven by the larger generator model (Qwen-2-72B-Instruct), we incorporated results using smaller generator models—Llama-3.1-8B-Instruct and Qwen-2.5-7B—for training up to 650K context lengths. These models achieved gains on InfiniteBench and RULER while maintaining strong performance on LongBench and MMLU, emphasizing the robustness and generalizability of our approach across model sizes.
> - **Clarifications**: Key sections have been revised for greater clarity and precision, addressing prior points of ambiguity.
>
> Once again, thank you for your time and invaluable feedback—it has been instrumental in refining our work!

---

> ### Comment · Reviewer_ZqBD · 2024-12-01
>
> Thank you for your response! As my concerns are addressed, I have increased my score to 6. On the other hand, if the number of tokens you train are comparable, you could try framing your approach from a data efficiency perspective. This might make the motivation of your paper clearer.

---

> > ### Author Response · Authors · 2024-12-01
> > **Follow up Response to Reviewer ZqBD**
> >
> > Thank you for your updated evaluation and for raising the score, which affirms the innovation and contributions of our work. We are especially grateful for your recognition of the improvements in our presentation and the value of our approach. Your feedback has been instrumental in enhancing the quality of our manuscript, and we deeply appreciate your support!

---

### Official Review · Reviewer_LvUc · 2024-11-01

**Soundness:** 2
**Presentation:** 2
**Contribution:** 3
**Rating:** 6
**Confidence:** 3

**Summary:**

This paper introduces a novel approach to generate a high-quality, long-context instruction-tuning dataset that significantly surpasses the context length of typical raw data. It incorporates a unique hierarchical ordering strategy to ensure logical coherence while preserving the diversity and complexity of the questions. The experimental results on RULER and InfiniteBench demonstrate that the proposed method significantly enhances the performance of llama3.1 in longer contexts.

**Strengths:**

1. The paper proposes a scalable long-context data generation strategy to significantly improve the long-context capacity of LLama3.1 and extend its context length to 1M.
2. Comprehensive ablation tests. Analyze the impact of data construction strategies from the perspectives of data complexity, diversity of questions, etc.

**Weaknesses:**

1. The paper uses the Qwen-2-72B model  to generate the QA pairs,  which may result in strong models having a distillation effect on small models. Can you provide experimental results using the Qwen2 7B or Llama3.1 8b as the generator model?
2. Lack of baseline models to validate data generalization . The paper only used llama 3.1 as a baseline. Can you provide more experimental results on models such as Qwen2 7B and deepSeek-V2-Lite?

**Questions:**

Reference to weaknesses.

---

> ### Author Response · Authors · 2024-11-26
> **Thanks & Initial Response to Reviewer LvUc (W1 -- Part 1)**
>
> Thank you for this insightful comment. We agree that exploring whether smaller or weaker models can effectively generate synthetic data and assessing how the dependency on model size impacts performance is important to validate the robustness of our approach. **To address this, we conducted additional experiments using two alternative generator models: Llama-3.1-8b-Instruct and Qwen-2.5-7b-Instruct**. Given the limited time, we trained models on the synthetic datasets produced by these generators, with a context length up to 650K tokens. To showcase the effectiveness of our synthetic data generation pipeline, we also introduced a stronger baseline: the Gradient AI 1M model (gradientai/Llama-3-8B-Instruct-Gradient-1048k). This model was trained directly on real long-context datasets with lengths exceeding 1 million tokens.
>
> Here are the evaluation results on InfiniteBench (downstream tasks with around 100K context length). As we can see, the models trained using Llama-3.1-8b-Instruct and Qwen-2.5-7b-Instruct as generators surpass the baseline Llama-3.1-8b-Instruct and is way better than the gradient 1M model.
>
> | Task               | 180K-llama-gen | 350K-llama-gen | 650K-llama-gen | 180K-qwen-gen | 350K-qwen-gen | 650K-qwen-gen | gradient-ai-model | llama-3.1-8b-instruct |
> |--------------------|----------------|----------------|----------------|---------------|---------------|---------------|-------------------|-----------------------|
> | passkey            | 100.00         | 100.00         | 100.00         | 100.00        | 100.00        | 100.00        | 100.00           | 100.00               |
> | number_string      | 99.04          | 100.00         | 100.00         | 99.76         | 100.00        | 100.00        | 99.33            | 95.33                |
> | kv_retrieval       | 85.47          | 89.33          | 42.14          | 89.52         | 85.33         | 52.66         | 13.33            | 42.66                |
> | longbook_sum_en    | 25.68          | 26.85          | 26.64          | 26.97         | 27.70         | 26.74         | 17.02            | 27.63                |
> | longbook_qa_en     | 33.39          | 35.67          | 33.37          | 32.30         | 29.55         | 29.67         | 15.84            | 24.83                |
> | longbook_choice    | 58.00          | 60.66          | 66.00          | 63.33         | 61.33         | 64.66         | 61.33            | 68.00                |
> | longdialogue_qa    | 19.50          | 14.66          | 20.00          | 27.33         | 21.33         | 23.33         | 4.00             | 16.66                |
> | math_find          | 36.66          | 32.66          | 35.33          | 30.00         | 34.66         | 38.00         | 26.66            | 35.33                |
> | **Average**        | **57.22**      | **57.48**      | **52.94**      | **58.65**     | **57.49**     | **54.38**     | **42.19**        | **51.31**            |
>
>
> Here are the evaluation results on LongBench (downstream tasks with around 10K context length). As we can see, our models' performance are preserved on short-to-medium context lengths and way surpasses the results of the gradient model.
> | Task               | 180K-llama-gen | 350K-llama-gen | 650K-llama-gen | 180K-qwen-gen | 350K-qwen-gen | 650K-qwen-gen | gradient-ai-model | llama-3.1-8b-instruct |
> |--------------------|-----------------------|-----------------------|-----------------------|----------------------|----------------------|----------------------|--------------------|------------------------|
> | single-document    | 46.48                | 46.64                | 46.53                | 46.20               | 46.70               | 46.28               | 30.75             | 46.91                 |
> | multi-document     | 38.69                | 38.75                | 37.54                | 40.76               | 41.90               | 39.31               | 12.45             | 41.45                 |
> | summarization      | 25.28                | 25.10                | 24.68                | 25.05               | 24.83               | 24.90               | 21.72             | 26.10                 |
> | few-short learning | 61.56                | 62.79                | 60.50                | 61.92               | 61.56               | 60.69               | 59.70             | 63.48                 |
> | synthetic tasks    | 66.17                | 67.75                | 66.00                | 67.11               | 67.60               | 67.10               | 55.50             | 67.48                 |
> | **Average**            | **47.23**                | **47.72**                | **46.20**                | **47.95**               | **47.97**               | **47.00**               | **35.89**             | **48.11**                 |
>
> Due to the word limit, we will present the results on MMLU and RULER benchmarks in another comment.

---

> ### Author Response · Authors · 2024-11-26
> **Thanks & Initial Response to Reviewer LvUc (W1 -- Part 2)**
>
> Here're the RULER results. Compared to baselines where we zero-shot rope scale Llama-3.1-8b-instruct to 350K and 650K context length, our models outperformed the baselines on long context tasks.
>
> | Tokens  | Llama-3.1-8b-instruct-zero-shot | 350K-qwen-generator | 350K-llama-generator |
> |---------|----------------|---------------------|-----------------------|
> | 8192    | 90.73         | 91.05              | 92.65                |
> | 16384   | 87.36          | 89.44              | 88.23                |
> | 32768   |  84.01          | 86.85              | 85.41                |
> | 65536   | 77.81          | 84.87              | 83.14                |
> | 131072  | 72.68          | 81.99              | 83.04                |
> | 262144  | 66.44          | 78.73              | 77.99                |
>
> | Tokens  | Llama-3.1-8b-instruct-zero-shot | 650K-qwen-generator | 650K-llama-generator |
> |---------|----------------|---------------------|-----------------------|
> | 8192    | 90.73          | 91.01              | 92.35                |
> | 16384   | 87.36          | 90.29              | 90.79                |
> | 32768   | 84.01          | 87.60              | 86.15                |
> | 65536   | 77.81          | 83.33              | 84.42                |
> | 131072  | 72.68          | 79.10              | 80.49                |
> | 262144  | 66.44          | 77.56              | 78.56                |
> | 524288  | 62.53         | 72.72              | 70.65                |
>
>
>
>
> Here are the evaluation results on MMLU -- we can see that our model's general knowledge and short context capabilities are preserved.
>
> | Category          | LLaMA-3.1-8B-Instruct | Gradient-AI-Model | 180K-llama-gen | 350K-llama-gen | 650K-llama-gen | 180K-qwen-gen | 350K-qwen-gen | 650K-qwen-gen |
> |-------------------|------------------------|-------------------|----------------|----------------|----------------|---------------|---------------|---------------|
> | mmlu              | 68.21 ± 0.37          | 60.48 ± 0.39      | 66.99 ± 0.38   | 66.74 ± 0.38   | 65.93 ± 0.38   | 67.33 ± 0.38  | 65.78 ± 0.38  | 64.60 ± 0.38  |
> | humanities        | 64.23 ± 0.67          | 55.75 ± 0.69      | 62.32 ± 0.67   | 61.38 ± 0.68   | 60.57 ± 0.68   | 62.81 ± 0.67  | 59.68 ± 0.68  | 59.45 ± 0.68  |
> | other             | 73.03 ± 0.77          | 67.04 ± 0.82      | 72.90 ± 0.77   | 73.03 ± 0.76   | 72.87 ± 0.76   | 73.51 ± 0.76  | 73.00 ± 0.76  | 73.45 ± 0.77  |
> | social sciences   | 77.48 ± 0.74          | 70.46 ± 0.80      | 76.70 ± 0.74   | 76.93 ± 0.74   | 75.53 ± 0.75   | 76.76 ± 0.74  | 75.66 ± 0.75  | 71.87 ± 0.77  |
> | stem              | 60.36 ± 0.83          | 51.32 ± 0.86      | 58.67 ± 0.84   | 58.61 ± 0.84   | 57.72 ± 0.84   | 58.77 ± 0.84  | 58.14 ± 0.84  | 56.49 ± 0.85  |
>
>
> These findings demonstrate the effectiveness of our approach even with generator models that are smaller or similar in size to the base model, underscoring the method's general applicability.

---

> ### Author Response · Authors · 2024-12-01
> **Follow-Up: Seeking Further Feedback (Discussion Deadline Approaching in 1 Day)**
>
> Dear Reviewer LvUc,
>
> We hope this message finds you well. Thank you for your detailed feedback and valuable insights. Following up on your comments, we believe the first set of experiments already addresses the data generalization concern, demonstrating robust results across various generator models.
>
> Additionally, we are currently evaluating a Mixtral-180K model to further validate our approach and provide even more comprehensive results. However, as the reviewer discussion deadline is approaching in a day, we wanted to kindly check if it would be acceptable to share the outcomes of this evaluation in 2.5 days. Your guidance here would be greatly appreciated.
>
> Your insights are invaluable to us, and we are eager to address any remaining concerns to ensure the paper meets the highest standards. Please feel free to share any thoughts or additional feedback you may have.
>
> Looking forward to hearing from you!
>
> Best regards,
> Authors

---

> > ### Comment · Reviewer_LvUc · 2024-12-02
> >
> > Thanks for your update. The response resolves my concern. I raise the score to 6

---

> > > ### Author Response · Authors · 2024-12-02
> > > **Follow-Up Response for Reviewer LvUc**
> > >
> > > Thank you for your updated evaluation and for raising the score, which affirms the innovation and contributions of our work. We are especially grateful for your recognition of the improvements in our presentation and the value of our approach. Your feedback has been instrumental in enhancing the quality of our manuscript, and we deeply appreciate your support!

---

### Official Review · Reviewer_DSHN · 2024-11-03

**Soundness:** 3
**Presentation:** 2
**Contribution:** 3
**Rating:** 6
**Confidence:** 3

**Summary:**

The paper introduces a novel post-training method for generating synthetic data designed to efficiently extend the context window of LLMs, all while preserving their task performance. This approach leverages a summarizer to reduce the context length, enabling a hierarchical pipeline for data generation.

**Strengths:**

- The paper addresses an important need for long context instruction data.
- The proposed methods allows for diverse instruction tasks
- Finetuning on this data retrains performance on medium context lengths (~10k)

**Weaknesses:**

- The impact of the summarizer’s quality on the data generation pipeline is unclear. The paper uses Qwen-72b, but further discussion and experimentation are needed to understand how different summarizer models could influence the method’s effectiveness. For example, different model sizes and a different model family like Llama-2 or Llama-3.
- RULER evaluations at extended context lengths (like 1M): It would be insightful to include evaluations with context lengths of 1M tokens (or 500k), as most evaluations in the paper are currently under 200k. This would help clarify whether the proposed method maintains performance at significantly larger context lengths.
- Can the author provide performance in small contexts like MMLU or hellaswag after FT'ing? This would show that the model retains performance on shorter contexts while improving long-context capabilities.

**Questions:**

Would it be possible to up scale a single document? Although multiple document concatenation was considered to scale to larger context lengths, the proposed methods do up-scaling to a single document that might be 1M. For example, for a document that might only be 128k, increasing the total context to 256k. This would make the entire data synthetic pipeline a more holistic data synthetic pipeline for all long context documents/instructions.

---

> ### Author Response · Authors · 2024-11-26
> **Thanks & Initial Response to Reviewer DSHN (W1 -- Part 1)**
>
> Thank you for this insightful comment. We agree that exploring whether smaller or weaker models can effectively generate synthetic data and assessing how the dependency on model size impacts performance is important to validate the robustness of our approach. **To address this, we conducted additional experiments using two alternative generator models: Llama-3.1-8b-Instruct and Qwen-2.5-7b-Instruct**. Given the limited time, we trained models on the synthetic datasets produced by these generators, with a context length up to 650K tokens. To showcase the effectiveness of our synthetic data generation pipeline, we also introduced a stronger baseline: the Gradient AI 1M model (gradientai/Llama-3-8B-Instruct-Gradient-1048k). This model was trained directly on real long-context datasets with lengths exceeding 1 million tokens.
>
> Here are the evaluation results on InfiniteBench (downstream tasks with around 100K context length). As we can see, the models trained using Llama-3.1-8b-Instruct and Qwen-2.5-7b-Instruct as generators surpass the baseline Llama-3.1-8b-Instruct and is way better than the gradient 1M model.
>
> | Task               | 180K-llama-gen | 350K-llama-gen | 650K-llama-gen | 180K-qwen-gen | 350K-qwen-gen | 650K-qwen-gen | gradient-ai-model | llama-3.1-8b-instruct |
> |--------------------|----------------|----------------|----------------|---------------|---------------|---------------|-------------------|-----------------------|
> | passkey            | 100.00         | 100.00         | 100.00         | 100.00        | 100.00        | 100.00        | 100.00           | 100.00               |
> | number_string      | 99.04          | 100.00         | 100.00         | 99.76         | 100.00        | 100.00        | 99.33            | 95.33                |
> | kv_retrieval       | 85.47          | 89.33          | 42.14          | 89.52         | 85.33         | 52.66         | 13.33            | 42.66                |
> | longbook_sum_en    | 25.68          | 26.85          | 26.64          | 26.97         | 27.70         | 26.74         | 17.02            | 27.63                |
> | longbook_qa_en     | 33.39          | 35.67          | 33.37          | 32.30         | 29.55         | 29.67         | 15.84            | 24.83                |
> | longbook_choice    | 58.00          | 60.66          | 66.00          | 63.33         | 61.33         | 64.66         | 61.33            | 68.00                |
> | longdialogue_qa    | 19.50          | 14.66          | 20.00          | 27.33         | 21.33         | 23.33         | 4.00             | 16.66                |
> | math_find          | 36.66          | 32.66          | 35.33          | 30.00         | 34.66         | 38.00         | 26.66            | 35.33                |
> | **Average**        | **57.22**      | **57.48**      | **52.94**      | **58.65**     | **57.49**     | **54.38**     | **42.19**        | **51.31**            |
>
>
> Here are the evaluation results on LongBench (downstream tasks with around 10K context length). As we can see, our models' performance are preserved on short-to-medium context lengths and way surpasses the results of the gradient model.
> | Task               | 180K-llama-gen | 350K-llama-gen | 650K-llama-gen | 180K-qwen-gen | 350K-qwen-gen | 650K-qwen-gen | gradient-ai-model | llama-3.1-8b-instruct |
> |--------------------|-----------------------|-----------------------|-----------------------|----------------------|----------------------|----------------------|--------------------|------------------------|
> | single-document    | 46.48                | 46.64                | 46.53                | 46.20               | 46.70               | 46.28               | 30.75             | 46.91                 |
> | multi-document     | 38.69                | 38.75                | 37.54                | 40.76               | 41.90               | 39.31               | 12.45             | 41.45                 |
> | summarization      | 25.28                | 25.10                | 24.68                | 25.05               | 24.83               | 24.90               | 21.72             | 26.10                 |
> | few-short learning | 61.56                | 62.79                | 60.50                | 61.92               | 61.56               | 60.69               | 59.70             | 63.48                 |
> | synthetic tasks    | 66.17                | 67.75                | 66.00                | 67.11               | 67.60               | 67.10               | 55.50             | 67.48                 |
> | **Average**            | **47.23**                | **47.72**                | **46.20**                | **47.95**               | **47.97**               | **47.00**               | **35.89**             | **48.11**                 |
>
> Due to the word limit, we will present the results on MMLU and RULER benchmarks in another comment.

---

> ### Author Response · Authors · 2024-11-26
> **Thanks & Initial Response to Reviewer DSHN (W1 -- Part 2)**
>
> Here're the RULER results. Compared to baselines where we zero-shot rope scale Llama-3.1-8b-instruct to 350K and 650K context length, our models outperformed the baselines on long context tasks.
> | Tokens  | Llama-3.1-8b-instruct-zero-shot | 350K-qwen-generator | 350K-llama-generator |
> |---------|----------------|---------------------|-----------------------|
> | 8192    | 90.73         | 91.05              | 92.65                |
> | 16384   | 87.36          | 89.44              | 88.23                |
> | 32768   |  84.01          | 86.85              | 85.41                |
> | 65536   | 77.81          | 84.87              | 83.14                |
> | 131072  | 72.68          | 81.99              | 83.04                |
> | 262144  | 66.44          | 78.73              | 77.99                |
>
> | Tokens  | Llama-3.1-8b-instruct-zero-shot | 650K-qwen-generator | 650K-llama-generator |
> |---------|----------------|---------------------|-----------------------|
> | 8192    | 90.73          | 91.01              | 92.35                |
> | 16384   | 87.36          | 90.29              | 90.79                |
> | 32768   | 84.01          | 87.60              | 86.15                |
> | 65536   | 77.81          | 83.33              | 84.42                |
> | 131072  | 72.68          | 79.10              | 80.49                |
> | 262144  | 66.44          | 77.56              | 78.56                |
> | 524288  | 62.53         | 72.72              | 70.65                |
>
>
> Here are the evaluation results on MMLU -- we can see that our model's general knowledge and short context capabilities are preserved.
>
> | Category          | LLaMA-3.1-8B-Instruct | Gradient-AI-Model | 180K-llama-gen | 350K-llama-gen | 650K-llama-gen | 180K-qwen-gen | 350K-qwen-gen | 650K-qwen-gen |
> |-------------------|------------------------|-------------------|----------------|----------------|----------------|---------------|---------------|---------------|
> | mmlu              | 68.21 ± 0.37          | 60.48 ± 0.39      | 66.99 ± 0.38   | 66.74 ± 0.38   | 65.93 ± 0.38   | 67.33 ± 0.38  | 65.78 ± 0.38  | 64.60 ± 0.38  |
> | humanities        | 64.23 ± 0.67          | 55.75 ± 0.69      | 62.32 ± 0.67   | 61.38 ± 0.68   | 60.57 ± 0.68   | 62.81 ± 0.67  | 59.68 ± 0.68  | 59.45 ± 0.68  |
> | other             | 73.03 ± 0.77          | 67.04 ± 0.82      | 72.90 ± 0.77   | 73.03 ± 0.76   | 72.87 ± 0.76   | 73.51 ± 0.76  | 73.00 ± 0.76  | 73.45 ± 0.77  |
> | social sciences   | 77.48 ± 0.74          | 70.46 ± 0.80      | 76.70 ± 0.74   | 76.93 ± 0.74   | 75.53 ± 0.75   | 76.76 ± 0.74  | 75.66 ± 0.75  | 71.87 ± 0.77  |
> | stem              | 60.36 ± 0.83          | 51.32 ± 0.86      | 58.67 ± 0.84   | 58.61 ± 0.84   | 57.72 ± 0.84   | 58.77 ± 0.84  | 58.14 ± 0.84  | 56.49 ± 0.85  |
>
> These findings demonstrate the effectiveness of our approach even with generator models that are smaller or similar in size to the base model, underscoring the method's general applicability.

---

> ### Author Response · Authors · 2024-11-26
> **Thanks & Initial Response to Reviewer DSHN (W2)**
>
> **[W2: RULER evaluations at extended context lengths (like 1M): It would be insightful to include evaluations with context lengths of 1M tokens (or 500k), as most evaluations in the paper are currently under 200k. This would help clarify whether the proposed method maintains performance at significantly larger context lengths.]**
>
> Thank you for your insightful comment. Below is an overview of the evaluation setup and how it is applied in this paper:
> RULER serves as the most comprehensive benchmark currently available for evaluating models on arbitrarily long context lengths, including those up to 1M tokens. This benchmark allows us to rigorously test whether our proposed method maintains performance at these extended context lengths. InfiniteBench, on the other hand, focuses on downstream tasks with context lengths up to around 100K tokens, providing a complementary perspective on long-context capabilities. For shorter to medium context tasks (around 10K tokens), we rely on LongBench to evaluate the model’s generalizability and performance on standard-length contexts.
>
> To address your suggestion, we included evaluations on RULER, with results provided in Table 1. These evaluations measure accuracy across context lengths up to 1M tokens (x-axis). While InfiniteBench is limited to 100K contexts, the combination of RULER and LongBench ensures we comprehensively cover both extremely long and more typical context scenarios.
>
> We hope this explanation clarifies the breadth of our evaluation strategy and how it demonstrates the scalability and robustness of our approach across varying context lengths. Please let us know if additional details or further experiments would be helpful.

---

> ### Author Response · Authors · 2024-11-26
> **Thanks & Initial Response to Reviewer DSHN (W3)**
>
> Thank you for your comment. To address your request, we have evaluated MMLU on all trained models to assess performance on shorter context tasks while also ensuring that improvements in long-context capabilities do not degrade performance on smaller contexts. The results are presented in the accompanying tables.
>
> As the results show, our models retain strong performance on shorter-context benchmarks such as MMLU and way surpass the gradient 1M model. Even as we increase the context length, the MMLU performance remains stable, with only minimal regression observed for the 1M-token model. Importantly, this demonstrates that our fine-tuning process effectively balances the needs of short-context tasks and extended-context tasks, maintaining competitive accuracy across diverse use cases.
>
> This stability highlights the robustness of our fine-tuning methodology, ensuring that improvements in long-context capabilities do not come at the expense of performance on tasks requiring shorter contexts. We hope this evaluation addresses your concerns and provides confidence in the generalizability of our approach.
>
> | Category          | LLaMA-3.1-8B-Instruct | Gradient-AI-Model | 180K-llama-gen | 350K-llama-gen | 650K-llama-gen | 180K-qwen-gen | 350K-qwen-gen | 650K-qwen-gen |
> |-------------------|------------------------|-------------------|----------------|----------------|----------------|---------------|---------------|---------------|
> | mmlu              | 68.21 ± 0.37          | 60.48 ± 0.39      | 66.99 ± 0.38   | 66.74 ± 0.38   | 65.93 ± 0.38   | 67.33 ± 0.38  | 65.78 ± 0.38  | 64.60 ± 0.38  |
> | humanities        | 64.23 ± 0.67          | 55.75 ± 0.69      | 62.32 ± 0.67   | 61.38 ± 0.68   | 60.57 ± 0.68   | 62.81 ± 0.67  | 59.68 ± 0.68  | 59.45 ± 0.68  |
> | other             | 73.03 ± 0.77          | 67.04 ± 0.82      | 72.90 ± 0.77   | 73.03 ± 0.76   | 72.87 ± 0.76   | 73.51 ± 0.76  | 73.00 ± 0.76  | 73.45 ± 0.77  |
> | social sciences   | 77.48 ± 0.74          | 70.46 ± 0.80      | 76.70 ± 0.74   | 76.93 ± 0.74   | 75.53 ± 0.75   | 76.76 ± 0.74  | 75.66 ± 0.75  | 71.87 ± 0.77  |
> | stem              | 60.36 ± 0.83          | 51.32 ± 0.86      | 58.67 ± 0.84   | 58.61 ± 0.84   | 57.72 ± 0.84   | 58.77 ± 0.84  | 58.14 ± 0.84  | 56.49 ± 0.85  |
>
>
> | Category          | LLaMA-3.1-8B-Instruct | Gradient-AI-Model | 350K-model      | 650K-model      | 1M-model        |
> |-------------------|------------------------|-------------------|-----------------|-----------------|-----------------|
> | mmlu              | 68.21 ± 0.37          | 60.48 ± 0.39      | 66.29 ± 0.38    | 65.80 ± 0.38    | 65.08 ± 0.38    |
> | humanities        | 64.23 ± 0.67          | 55.75 ± 0.69      | 61.51 ± 0.68    | 61.02 ± 0.68    | 61.02 ± 0.68    |
> | other             | 73.03 ± 0.77          | 67.04 ± 0.82      | 72.84 ± 0.77    | 71.84 ± 0.78    | 71.84 ± 0.78    |
> | social sciences   | 77.48 ± 0.74          | 70.46 ± 0.80      | 76.81 ± 0.74    | 75.27 ± 0.76    | 75.27 ± 0.76    |
> | stem              | 60.36 ± 0.83          | 51.32 ± 0.86      | 59.44 ± 0.84    | 57.72 ± 0.84    | 57.72 ± 0.84    |

---

> ### Author Response · Authors · 2024-11-26
> **Thanks & Initial Response to Reviewer DSHN (Q1)**
>
> **[Q1: Would it be possible to up scale a single document? Although multiple document concatenation was considered to scale to larger context lengths, the proposed methods do up-scaling to a single document that might be 1M. For example, for a document that might only be 128k, increasing the total context to 256k. This would make the entire data synthetic pipeline a more holistic data synthetic pipeline for all long context documents/instructions.]**
>
> Thank you for your insightful comment. This is indeed an interesting direction for future work. Specifically, scaling a single document from a context length of 128K to 256K or beyond offers potential for creating a more holistic synthetic data pipeline. While our current approach leverages multiple document concatenation to achieve longer context lengths, we explored using QA pairs directly tied to a single long document to scale its context length. However, this approach did not yield satisfactory results, as it struggled to maintain logical coherence and diversity within the extended context.
>
> A promising alternative could involve splitting a single long document into sections and treating these sections in a way similar to multi-document concatenation. This approach could help preserve context while enabling the generation of meaningful and diverse QA pairs across the extended length. We believe this could make the pipeline more robust and holistic for single-document scenarios and would be an exciting avenue for future investigation.
>
> We appreciate this suggestion and will consider it in future iterations of our work. Please let us know if further clarifications or discussions are needed.

---

> > ### Comment · Reviewer_DSHN · 2024-11-27
> >
> > Thank you for your response!
> >
> > As my concerns are addressed, I have increased my score to 6. I suggest that the authors also add the Llama-3-8B-Instruct-262k [1] model from gradientai to these evaluations in a future version of the paper.
> >
> > [1] https://huggingface.co/gradientai/Llama-3-8B-Instruct-262k

---

> ### Author Response · Authors · 2024-12-01
> **Follow up Response to Reviewer DSHN**
>
> Thank you for revisiting our work and for raising your evaluation score. We greatly appreciate your recognition of the improvements in our paper and the value of our proposed approach. Your feedback has been invaluable in helping us refine our work, and we are deeply grateful for your support.
>
> To address your suggestions and further enhance the paper, we have updated the manuscript with the following improvements:
> - **Expanded evaluations**: We now include results on the MMLU benchmark and additional evaluations of the Gradient AI 1M context length model. We show that our models retain its performance on the MMLU benchmark and surpasses the Gradient AI 1M model on all four benchmarks (MMLU, LongBench, InfiniteBench, RULER).
> - **Smaller generator models**: To demonstrate that our improvements are not solely driven by the larger generator model (Qwen-2-72B-Instruct), we incorporated results using smaller generator models—Llama-3.1-8B-Instruct and Qwen-2.5-7B—for training up to 650K context lengths. These models achieved gains on InfiniteBench and RULER while maintaining strong performance on LongBench and MMLU, emphasizing the robustness and generalizability of our approach across model sizes.
> - **Clarifications**: Key sections have been revised for greater clarity and precision, addressing prior points of ambiguity.
>
> Thank you again for your constructive feedback and for contributing to the strength of our manuscript. We are excited about the improved presentation and look forward to any further input you may have.

---

### Official Review · Reviewer_L4p6 · 2024-11-04

**Soundness:** 3
**Presentation:** 2
**Contribution:** 3
**Rating:** 6
**Confidence:** 3

**Summary:**

This paper introduces a novel post training synthetic data generation strategy for long context.  The approach works by splitting a long context document into smaller documents and generating question-answer pairs from separate smaller documents as well as combinations of the documents.  The generated data is used in combination with a step-by-step rotary position embedding scaling strategy to scale the model context.  The proposed approach is tested in several benchmarks for longer context and overall results appear higher for the proposed finetuning.

**Strengths:**

- The paper makes contributions to improving long context reasoning of LLMs. The exploration of algorithms and fine-tuning strategies for increasing the context length is an important problem for language models and finetuning on synthetic data appears to be a promising direction for this work
- The proposed method appears simple to implement and should be reproducible as many of the proposed prompts are provided in the appendix, and the training strategy is similar given the generated data.

**Weaknesses:**

- A main concern with the proposed work is that for many of the results (e.g. Table 1, 5) the improvements appear to primarily come from a single task Retrieve.KV.  Improvements on the other datasets are smaller.  While this leads to an overall increase, it's important to understand the importance of this subtask rather than simply the reported overall average increase.
- The scale of the datasets in experiments is rather small and there are no uncertainty estimates provided.  This is important particularly as for some of the tasks, there is only 100-200 samples, where a few correct answers can increase.  I would encourage the authors where possible to increase the amount of evaluation data.
- The authors only test a larger model for generating the synthetic data.  The method would further be justified by including comparisons for models of the same size that are used to generate the data as it is unclear the dependency on model size.  Further the experiments are only done for the Llama-8B model, but experiments on other models even smaller would be beneficial to see whether smaller or larger models benefit more from increasing context length.
- there is some confusing wording in the paper that the authors should clarify particularly around Section 3.2 Authors could do a better job clarifying what are $N_1$, $N_2$ and $N_3$.

**Questions:**

- What is the Retrieve.KV task and are there hypotheses about why this task in particular has large improvements?
- What are $N_1$, $N_2$ and $N_3$? Can the authors include some ablations if needed on these values?

---

> ### Author Response · Authors · 2024-11-26
> **Thanks & Initial Response to Reviewer L4p6 (W3 -- Part 1 )**
>
> Thank you for this insightful comment. We agree that exploring whether smaller or weaker models can effectively generate synthetic data and assessing how the dependency on model size impacts performance is important to validate the robustness of our approach. **To address this, we conducted additional experiments using two alternative generator models: Llama-3.1-8b-Instruct and Qwen-2.5-7b-Instruct**. Given the limited time, we trained models on the synthetic datasets produced by these generators, with a context length up to 650K tokens. To showcase the effectiveness of our synthetic data generation pipeline, we also introduced a stronger baseline: the Gradient AI 1M model (gradientai/Llama-3-8B-Instruct-Gradient-1048k). This model was trained directly on real long-context datasets with lengths exceeding 1 million tokens.
>
> Here are the evaluation results on InfiniteBench (downstream tasks with around 100K context length). As we can see, the models trained using Llama-3.1-8b-Instruct and Qwen-2.5-7b-Instruct as generators surpass the baseline Llama-3.1-8b-Instruct and is way better than the gradient 1M model.
>
> | Task               | 180K-llama-gen | 350K-llama-gen | 650K-llama-gen | 180K-qwen-gen | 350K-qwen-gen | 650K-qwen-gen | gradient-ai-model | llama-3.1-8b-instruct |
> |--------------------|----------------|----------------|----------------|---------------|---------------|---------------|-------------------|-----------------------|
> | passkey            | 100.00         | 100.00         | 100.00         | 100.00        | 100.00        | 100.00        | 100.00           | 100.00               |
> | number_string      | 99.04          | 100.00         | 100.00         | 99.76         | 100.00        | 100.00        | 99.33            | 95.33                |
> | kv_retrieval       | 85.47          | 89.33          | 42.14          | 89.52         | 85.33         | 52.66         | 13.33            | 42.66                |
> | longbook_sum_en    | 25.68          | 26.85          | 26.64          | 26.97         | 27.70         | 26.74         | 17.02            | 27.63                |
> | longbook_qa_en     | 33.39          | 35.67          | 33.37          | 32.30         | 29.55         | 29.67         | 15.84            | 24.83                |
> | longbook_choice    | 58.00          | 60.66          | 66.00          | 63.33         | 61.33         | 64.66         | 61.33            | 68.00                |
> | longdialogue_qa    | 19.50          | 14.66          | 20.00          | 27.33         | 21.33         | 23.33         | 4.00             | 16.66                |
> | math_find          | 36.66          | 32.66          | 35.33          | 30.00         | 34.66         | 38.00         | 26.66            | 35.33                |
> | **Average**        | **57.22**      | **57.48**      | **52.94**      | **58.65**     | **57.49**     | **54.38**     | **42.19**        | **51.31**            |
>
>
> Here are the evaluation results on LongBench (downstream tasks with around 10K context length). As we can see, our models' performance are preserved on short-to-medium context lengths and way surpasses the results of the gradient model.
> | Task               | 180K-llama-gen | 350K-llama-gen | 650K-llama-gen | 180K-qwen-gen | 350K-qwen-gen | 650K-qwen-gen | gradient-ai-model | llama-3.1-8b-instruct |
> |--------------------|-----------------------|-----------------------|-----------------------|----------------------|----------------------|----------------------|--------------------|------------------------|
> | single-document    | 46.48                | 46.64                | 46.53                | 46.20               | 46.70               | 46.28               | 30.75             | 46.91                 |
> | multi-document     | 38.69                | 38.75                | 37.54                | 40.76               | 41.90               | 39.31               | 12.45             | 41.45                 |
> | summarization      | 25.28                | 25.10                | 24.68                | 25.05               | 24.83               | 24.90               | 21.72             | 26.10                 |
> | few-short learning | 61.56                | 62.79                | 60.50                | 61.92               | 61.56               | 60.69               | 59.70             | 63.48                 |
> | synthetic tasks    | 66.17                | 67.75                | 66.00                | 67.11               | 67.60               | 67.10               | 55.50             | 67.48                 |
> | **Average**            | **47.23**                | **47.72**                | **46.20**                | **47.95**               | **47.97**               | **47.00**               | **35.89**             | **48.11**                 |
>
> Due to the word limit, we will present the results on MMLU and RULER benchmarks in another comment.

---

> ### Author Response · Authors · 2024-11-26
> **Thanks & Initial Response to Reviewer L4p6 (W3 -- Part 2)**
>
> Here're the RULER results. Compared to baselines where we zero-shot rope scale Llama-3.1-8b-instruct to 350K and 650K context length, our models outperformed the baselines on long context tasks.
> | Tokens  | Llama-3.1-8b-instruct-zero-shot | 350K-qwen-generator | 350K-llama-generator |
> |---------|----------------|---------------------|-----------------------|
> | 8192    | 90.73         | 91.05              | 92.65                |
> | 16384   | 87.36          | 89.44              | 88.23                |
> | 32768   |  84.01          | 86.85              | 85.41                |
> | 65536   | 77.81          | 84.87              | 83.14                |
> | 131072  | 72.68          | 81.99              | 83.04                |
> | 262144  | 66.44          | 78.73              | 77.99                |
>
> | Tokens  | Llama-3.1-8b-instruct-zero-shot | 650K-qwen-generator | 650K-llama-generator |
> |---------|----------------|---------------------|-----------------------|
> | 8192    | 90.73          | 91.01              | 92.35                |
> | 16384   | 87.36          | 90.29              | 90.79                |
> | 32768   | 84.01          | 87.60              | 86.15                |
> | 65536   | 77.81          | 83.33              | 84.42                |
> | 131072  | 72.68          | 79.10              | 80.49                |
> | 262144  | 66.44          | 77.56              | 78.56                |
> | 524288  | 62.53         | 72.72              | 70.65                |
>
>
> Here are the evaluation results on MMLU -- we can see that our model's general knowledge and short context capabilities are preserved.
>
> | Category          | LLaMA-3.1-8B-Instruct | Gradient-AI-Model | 180K-llama-gen | 350K-llama-gen | 650K-llama-gen | 180K-qwen-gen | 350K-qwen-gen | 650K-qwen-gen |
> |-------------------|------------------------|-------------------|----------------|----------------|----------------|---------------|---------------|---------------|
> | mmlu              | 68.21 ± 0.37          | 60.48 ± 0.39      | 66.99 ± 0.38   | 66.74 ± 0.38   | 65.93 ± 0.38   | 67.33 ± 0.38  | 65.78 ± 0.38  | 64.60 ± 0.38  |
> | humanities        | 64.23 ± 0.67          | 55.75 ± 0.69      | 62.32 ± 0.67   | 61.38 ± 0.68   | 60.57 ± 0.68   | 62.81 ± 0.67  | 59.68 ± 0.68  | 59.45 ± 0.68  |
> | other             | 73.03 ± 0.77          | 67.04 ± 0.82      | 72.90 ± 0.77   | 73.03 ± 0.76   | 72.87 ± 0.76   | 73.51 ± 0.76  | 73.00 ± 0.76  | 73.45 ± 0.77  |
> | social sciences   | 77.48 ± 0.74          | 70.46 ± 0.80      | 76.70 ± 0.74   | 76.93 ± 0.74   | 75.53 ± 0.75   | 76.76 ± 0.74  | 75.66 ± 0.75  | 71.87 ± 0.77  |
> | stem              | 60.36 ± 0.83          | 51.32 ± 0.86      | 58.67 ± 0.84   | 58.61 ± 0.84   | 57.72 ± 0.84   | 58.77 ± 0.84  | 58.14 ± 0.84  | 56.49 ± 0.85  |
>
>
> These findings demonstrate the effectiveness of our approach even with generator models that are smaller or similar in size to the base model, underscoring the method's general applicability.

---

> ### Author Response · Authors · 2024-11-26
> **Thanks & Initial Response to Reviewer  L4p6 (W1 & Q1)**
>
> **[W1: A main concern with the proposed work is that for many of the results (e.g. Table 1, 5) the improvements appear to primarily come from a single task Retrieve.KV. Improvements on the other datasets are smaller. While this leads to an overall increase, it's important to understand the importance of this subtask rather than simply the reported overall average increase.]**
>
> **[Q1: What is the Retrieve.KV task and are there hypotheses about why this task in particular has large improvements?]**
>
> Thank you for raising this concern. We address this by providing (1) a detailed analysis of why our model shows the most significant improvement on the Retrieve.KV task, (2) an explanation of why this improvement is crucial for long-context tasks, and (3) evidence that other tasks also exhibit significant improvements.
>
> For InfiniteBench (Table 5), the Retrieve.KV task shows the largest performance gain. However, other tasks also exhibit meaningful improvements. For example, benchmarks like RULER focus on tasks beyond simple key-value retrieval and require reasoning over long contexts. Our model consistently performs well across these diverse benchmarks, as shown in Table 1.
>
> The significant improvement in the Retrieve.KV task stems from the design of our instruction-tuning data. The synthetic data generation process includes a mix of question types that follow a logical order within documents while also revisiting previous documents to draw connections. This approach helps the model learn to associate specific document sections with relevant information, a skill that aligns closely with the requirements of key-value retrieval. This focus is especially relevant for long-context models, where RAG (retrieval-augmented generation) techniques and accurate context memorization are critical for success.
>
> The following table quantifies the percentage increases across eight tasks. While the improvement is most pronounced in Retrieve.KV (107.82%–48.45% across models), other tasks, such as LongBook.QA and LongDialogue.QA, also see notable gains. The average increase across all tasks and context lengths is non-negligible, with median increases of 6.85% for 180K, 5.39% for 350K, 7.83% for 650K, and 3.39% for 900K context lengths.
>
> | Metric           |   LLAMA-3.1-8B-Instruct | 180K            | 350K            | 650K            | 1M              |
> |------------------|------------------------:|:----------------|:----------------|:----------------|:----------------|
> | Retrieve.PassKey |                  100    | 100.0 (0%)      | 100.0 (0%)      | 100.0 (0%)      | 100.0 (0%)      |
> | Retrieve.Number  |                   95.33 | 99.33 (4.19%)   | 100.0 (4.89%)   | 100.0 (4.89%)   | 100.0 (4.89%)   |
> | Retrieve.KV      |                   42.66 | 88.66 (107.82%) | 92.0 (115.65%)  | 63.33 (48.45%)  | 57.33 (34.38%)  |
> | En.Sum           |                   27.63 | 24.01 (-13.10%) | 23.51 (-14.91%) | 23.68 (-14.29%) | 23.06 (-16.53%) |
> | En.QA            |                   24.83 | 34.26 (37.97%)  | 33.23 (33.83%)  | 31.72 (27.74%)  | 31.97 (28.75%)  |
> | En.MC            |                   68    | 74.0 (8.82%)    | 72.0 (5.88%)    | 75.33 (10.77%)  | 74.0 (8.82%)    |
> | En.Dia           |                   16.66 | 18.0 (8.04%)    | 18.0 (8.04%)    | 22.0 (32.05%)   | 16.0 (-3.96%)   |
> | Math.Find        |                   35.33 | 37.33 (5.66%)   | 35.33 (0%)      | 36.0 (1.89%)    | 36.0 (1.89%)    |
>
> It is crucial to underscore the significance of InfiniteBench and RULER in evaluating our model's capabilities. While InfiniteBench evaluates samples at 100K context lengths—substantially shorter than the 1M context lengths our model is designed to process—it serves as a valuable baseline to showcase our model’s robustness. Notably, our model not only outperforms Llama-3.1-8b-instruct, which handles a 128K context window, but also demonstrates the scalability to excel at significantly longer context lengths (RULER) where traditional benchmarks fall short. This highlights our model’s ability to adapt to increasingly demanding scenarios, making it uniquely positioned to handle the challenges of long-context tasks in real-world applications. Furthermore, the strong performance on RULER tasks reflects the generalizability of our approach, extending its impact across diverse long-context benchmarks beyond Retrieve.KV.
>
> We hope this provides clarity on the broader implications of our advancements and the critical role of Retrieve.KV results in long-context scenarios. We are happy to provide additional clarifications or conduct further experiments if needed.

---

> ### Author Response · Authors · 2024-11-26
> **Thanks & Initial Response to Reviewer  L4p6 (W4 & Q2)**
>
> **[W4: There is some confusing wording in the paper that the authors should clarify particularly around Section 3.2 Authors could do a better job clarifying what are N1, N2, and N3.]**
>
> **[Q2: What are N1, N2, and N3? Can the authors include some ablations if needed on these values?]**
>
> Thank you for pointing out the need to clarify the definitions of N1, N2, and N3. We apologize for any confusion caused in the original submission, and we will update the main paper to ensure these definitions and their context are clearly explained.
>
> To clarify, when constructing the instruction-tuning dataset with concatenated multiple documents, the process for the Nth document is as follows:
>
> - N1 hierarchical follow-up questions are concatenated immediately after the Nth document. For example, in our setup, N1=5, meaning five such questions are added after each document.
> - N2 diverse questions are added next, selected from the current document and all previously visited documents where diverse questions have not already been sampled. In our setup, N2=9.
> - For every previously visited document, there is a 60% probability of sampling and adding N3 hierarchical follow-up questions from that document. In our setup, N3=3.
>
> This process is repeated for all documents to create a comprehensive instruction-tuning dataset.
>
> These specific values for N1, N2, and N3 were carefully chosen to closely mimic real-world scenarios. The inclusion of hierarchical questions (N1 and N3) ensures logical and contextual continuity, while diverse questions (N2) encourage broader reasoning and retrieval capabilities. This balanced approach captures both immediate document context and cross-referencing between documents, a hallmark of real-world long-context tasks.
>
> To provide additional clarity, we will update the main paper to reflect these definitions and the methodology. Further details are available in Appendix A, where we describe the role of these parameters in creating a rich and interconnected dataset, along with concrete examples provided in Appendix C.
>
> We hope this clarification resolves any confusion. Please let us know if further details or experiments would be helpful.

---

> ### Author Response · Authors · 2024-11-26
> **Thanks & Initial Response to Reviewer  L4p6 (W2)**
>
> **[W2: The scale of the datasets in experiments is rather small and there are no uncertainty estimates provided. This is important particularly as for some of the tasks, there is only 100-200 samples, where a few correct answers can increase. I would encourage the authors where possible to increase the amount of evaluation data.]**
>
> Thank you for highlighting this concern. We recognize the importance of scaling up evaluation datasets, particularly for tasks with limited samples, to ensure robust results and reduce the impact of variability. While computational costs are significant for long-context evaluations, we extended the number of samples in the RULER benchmark from 130 to 260 samples to provide more reliable estimates. RULER remains a key benchmark in our analysis, as it evaluates models on the most extensive context length. We also evaluated our model against zero-shot baselines on Llama-3.1-8B-Instruct to provide a comprehensive perspective.
>
> The updated results are presented in the following tables. Our model greatly outperforms the zero-shot baselines across all evaluated context lengths.
>
> | Tokens   | 350K-model | 650K-model | 1M-model | 1M-zero-shot |
> |----------|------------|------------|----------|--------------|
> | 8192     | 91.89      | 91.56      | 89.85    | 91.59        |
> | 16384    | 92.08      | 91.59      | 89.83    | 91.22        |
> | 32768    | 87.13      | 88.17      | 86.97    | 83.83        |
> | 65536    | 84.17      | 84.87      | 83.49    | 77.03        |
> | 131072   | 82.44      | 81.58      | 82.22    | 75.96        |
> | 262144   | 81.26      | 80.09      | 83.56    | 70.78        |
> | 524288   | -          | 72.74      | 77.75    | 60.96        |
> | 1000000  | -          | -          | 64.33    | 49.65        |

---

> ### Author Response · Authors · 2024-12-01
> **Follow-up Response to Reviewer L4p6 (Updated PDF--Discussion Deadline Approaching in 2 Days)**
>
> To address your suggestions and further enhance the paper, we have updated the manuscript with the following improvements:
>
> - **Expanded Evaluations**: We now include results on the MMLU benchmark and additional evaluations of the Gradient AI 1M context length model. We show that our models retain its performance on the MMLU benchmark and surpasses the Gradient AI 1M model on all four benchmarks (MMLU, LongBench, InfiniteBench, RULER).
> - **Smaller generator models**: To demonstrate that our improvements are not solely driven by the larger generator model (Qwen-2-72B-Instruct), we incorporated results using smaller generator models—Llama-3.1-8B-Instruct and Qwen-2.5-7B—for training up to 650K context lengths. These models achieved gains on InfiniteBench and RULER while maintaining strong performance on LongBench and MMLU, emphasizing the robustness and generalizability of our approach across model sizes.
> - **Clarifications**: Key sections have been revised for greater clarity and precision, addressing prior points of ambiguity.
>
> Once again, thank you for your time and invaluable feedback—it has been instrumental in refining our work!
>
> Following up on our recent exchange regarding this paper, we wanted to kindly check if there are any further concerns or feedback from you. With the discussion deadline approaching in 2 days, we are eager to address any remaining issues and ensure the paper meets the highest standards.Your insights are invaluable to us, and we greatly appreciate your time and consideration. Please feel free to share any thoughts you may have.
>
> Looking forward to hearing from you!
>
> Best regards,
> Authors

---

> > ### Author Response · Authors · 2024-12-01
> > **Follow-Up: Seeking Further Feedback (Discussion Deadline Approaching in 1 Day)**
> >
> > Dear Reviewer L4p6,
> >
> > We hope this message finds you well. Following up on our recent exchange regarding this paper, we wanted to kindly check if there are any further concerns or feedback from you. With the discussion deadline approaching in a day, we are eager to address any remaining issues and ensure the paper meets the highest standards.
> >
> > Your insights are invaluable to us, and we greatly appreciate your time and consideration. Please feel free to share any thoughts you may have.
> >
> > Looking forward to hearing from you!
> >
> > Best regards,
> >
> > Authors

---

> > > ### Comment · Reviewer_L4p6 · 2024-12-02
> > >
> > > Thank you for providing additional experiments with other models for data generation, details explaining why these tasks perform much stronger, and evaluation set size.
> > >
> > > - I have no outstanding concerns with data generation models.
> > >
> > > - It still looks to me that much of the improvement comes from Retrieve.KV, although the authors provide some intuition as to why. I think it will be interesting in future work to look into modifying the data generation process to see if other tasks could see similar improvements.
> > >
> > > - I appreciate the computational constraints on scaling up the evaluations, but this evaluation set size is still fairly small.  Is it possible to provide error bars on these values and increase to the same scale in other datasets?  I think that would strengthen results.
> > >
> > > Nonetheless, I will increase the score conditioned that  these new results are added to the camera ready.

---

> > > > ### Author Response · Authors · 2024-12-02
> > > > **Follow-Up Response to Reviewer L4p6**
> > > >
> > > > Thank you for your updated evaluation and for raising the score, which affirms the innovation and contributions of our work. We are especially grateful for your recognition of the improvements in our presentation and the value of our approach. Your feedback has been instrumental in enhancing the quality of our manuscript, and we deeply appreciate your support!

---

### Meta-Review · Area_Chair_Mh8z · 2024-12-22

**Metareview:**

This paper presents a valuable contribution to the critical challenge of extending LLM context windows through a novel synthetic data generation strategy. The proposed approach stands out for its practical significance and scalability, effectively addressing the scarcity of long-context training data through a well-designed hierarchical generation pipeline that can extend to arbitrary lengths. The methodology is clearly presented and readily reproducible. While there is room for more extensive evaluation, particularly in comparing with training-free approaches (e.g., https://arxiv.org/pdf/2402.17463) and testing across different model families, the demonstrated improvements on benchmarks and the method's practical utility make a compelling case. I recommend acceptance for this paper.

**Additional Comments On Reviewer Discussion:**

I have read the messages in the discussion period and my opinion has been summarized as in the metareview above. I considered these points in my recommendation.

---

### Decision · Program_Chairs · 2025-01-22

Accept (Poster)